# Development and preliminary validation of the GebStart-tool for advising nulliparous women in early labour

Susanne Grylka-Baeschlin[1]*, Nadine Pauli[2], Catherine Rapp[3], Carola Baumgartner[4], Clizia Iseppi[5], Nele Struebing[6], Linda Karg[7], Gabriela Minati[7], Leonhard Schäffer[2], Olav Lapaire[3], Markus Hodel[4], Gabriella Stocker[5], Nina Kimmich[6], Leila Sultan-Beyer[7], Antonia Nathalie Mueller[1]

1 Research Institute of Midwifery and Reproductive Health, School of Health Sciences, ZHAW Zurich University of Applied Sciences, Winterthur, Switzerland, 2 Division of Obstetrics and Prenatal Diagnostics, Cantonal Hospital of Baden, Baden, Switzerland, 3 University Hospital of Basel, Women's Clinic, Basel, Switzerland, 4 Cantonal Hospital of Lucerne, Women's Hospital, Lucerne, Switzerland, 5 Stadtspital Zürich Triemli, Gynaecology and Maternité, Zurich, Switzerland, 6 University Hospital Zurich, Clinic for Obstetrics, Zurich, Switzerland, 7 Cantonal Hospital of Winterthur, Women's Hospital, Winterthur, Switzerland

* susanne.grylka@zhaw.ch

## Abstract

### Objectives

Nulliparous women in early labour are unsure when to go to hospital. The aim of this study was to develop and preliminary validate a tool for advising for or against hospital admission.

### Methods

We developed the preliminary long version of the GebStart-tool with 32 items based on focus group discussions and a scoping review. It was applied in a multicentre study with n = 394 women during their contact with the hospital. Because of the formative and complex character of the GebStart-tool, factor analysis was not appropriate. Instead, items were subdivided deductively into the domains 'Physical symptoms', 'Emotional state', Self-management' and 'Resources'. Distribution of response options, adjusted Cox regressions with time intervals describing care needs as outcomes and adjusted multinomial regression with the outcome 'Care decision' were used to reduce items and for preliminary validation.

### Results

The reduced GebStart-tool contained 15 items and cutoff points at 22 and 33 points. The total score of the instrument was significantly associated with all time intervals describing care needs (duration between completion of the tool and hospital admission (HR = 1.08, 95% CI [1.05–1.10], p < 0.001), onset of active labour (HR = 1.06, 95%

**Data availability statement:** All relevant data for this study are publicly available from the Zenodo repository (https://doi.org/10.5281/zenodo.12773346).

**Funding:** This study was funded by the Swiss National Science Foundation with a Practice-to-Science Grant (PT00P1_199085). The funders had no role in study design, data collection and analysis, decision to publish, or preparation of the manuscript.

**Competing interests:** The authors have declared that no competing interests exist.

CI [1.04–1.08], p < 0.001), first use of medical pain management (HR = 1.08, 95% CI [1.06–1.11], p < 0.001), first use of alternative pain management (HR = 1.08, 95% CI [1.05–1.10], p < 0.001)). However, a higher total score of the reduced GebStart-tool was not significantly associated with a reduced risk for the decision 'Stay at home' (RR = 0.98, 95% CI [0.94–1.02], p = 0.421), but with a significantly higher risk for the decision 'Hospital admission' (RR = 1.13, 95% CI [1.05–1.22], p = 0.001) compared to 'Keep in contact'.

## Conclusion

We developed a practical instrument with 15 items based on scientific evidence. Further research of the GebStart-tool in larger samples is necessary. Moreover, the use in clinical practice accompanied by implementation research and translation into other languages should be envisaged.

## Introduction

The first contact of nulliparous women at the onset of labour with their intrapartum care providers may be crucial for the subsequent labour and birth process. Pregnant women expect midwifery support and individualised care during early labour [1]. Therefore, assessing women's personal needs might have the potential to improve woman-centred early labour care.

A heterogenous definition of the onset of labour and a sometimes unclear distinction between early and active labour are reasons for medically unjustified interventions early in the labour and birth process [2–4]. Following the onset of labour, the latent phase of labour or early labour together with the subsequent active phase comprise the first stage of labour [5]. Previous studies have shown that early interventions are especially common if women are admitted to hospital before active labour starts [6–8]. An increased risk of caesarean birth, labour augmentation with oxytocin, artificial rupture of the membranes and epidural analgesia were observed in association with an early hospital admission [6,7,9]. For these reasons, pregnant women are advised to stay at home as long as possible and midwives act as gatekeepers to delay hospital admission [10]. However, especially in times of high workload and financial pressure, advices might depend on the circumstances and can be arbitrary [11,12]. In addition, staying at home over 24 hours in women with prolonged early labour was also found to be associated with negative birth outcomes and an increased risk for caesarean section [13]. Therefore, this well-intended act to delay hospital admission and protect women from intrapartum interventions may lead to their needs not being met [10].

In this context, some pregnant women are unsatisfied with early labour care, especially nulliparae with prolonged early labour [1]. They and their labour companions have difficulties coping at home without professional support [1,10]. This particularly occurs if women have prolonged latent phases or are affected by fear of childbirth because fear can increase pain intensity [1,14]. A systematic review including five

trials has shown that standardised interventions during early labour such as home visits or using an algorithm for labour diagnosis increased women's satisfaction and had the potential to reduce intrapartum interventions, but did not have a clear impact on the mode of birth [15]. The results of this study suggest that the very individual character of labour onset and early labour would require more woman-centred care during this frequently difficult labour phase. Providing satisfying early labour care is therefore very challenging and consequently also frustrating for health professionals [16]. There is a lack of tools to assist early labour assessment in an evidence-based way to avoid arbitrary recommendations. Hence, we hypothesised that an evidence-based tool providing a holistic view of women's physical and mental states during early labour and indicating the individual needs of nulliparae is urgently needed to improve care during this vulnerable labour phase [17].

This study therefore aimed to develop and preliminary validate a tool for advising nulliparous women during early labour for or against hospital admission.

## Materials and methods

### Study design and setting

We conducted an instrument development study using a multistep approach [17,18]. Following an evidence-based development, the preliminary long version of the tool was applied in six Swiss hospitals for item reduction and preliminary validation. This multicentre data collection took place between March 1, 2022 and August 31, 2023 [17]. In addition, the study was registered in the Swiss National Clinical Trials Portal (SNCTP000004555, 27 July 2021) and the German Clinical Trial Register (DRKS00025572, 28 July 2021). The STrengthening the Reporting of OBservational studies in Epidemiology (STROBE) guidelines were followed for reporting [19] (S1 Table).

### Participants

Pregnant women ≥ 18 years old, expecting their first child, with a singleton in cephalic presentation and who did not plan either an elective caesarean birth or a labour induction, who had sufficient German knowledge, and who gave birth in one of the six study centres, were eligible for participation. For this development study it seemed appropriate to focus on nulliparae, because early labour duration and outcomes differ largely from those of multiparae [6,20].

A total of n = 627 women were recruited in the main study phase. We calculated the required sample size based on the initially planned exploratory factor analysis envisaging approximately n = 400 women with complete data for an estimated number of items of max. 40 and the common rule of ten participants per item [21]. As the tool contained 32 items, the minimum acceptable sample size was n = 320 women. It was estimated prior to the recruitment that approximately 25% of recruited women might excluded before the application of the tool because of unexpected caesarean birth or medically induced labour without spontaneous labour onset. The target sample size for the recruitment was then rounded up to n = 550 participants. Recruitment and dropout rates were both higher than expected leading eventually to an adequate sample size of n = 394 participants with at least one completed tool.

### Development of the GebStart-tool and data collection instruments

The detailed development process of the GebStart-tool was previously described in the published study protocol [17]. In brief, a pool of 99 items was developed based on results of a scoping review with an extensive literature search [22,23] and on focus group discussions with n = 18 nulliparous women [24,25]. Content and face validity assessed by n = 8 experts from Switzerland, Germany and Austria informed item reduction to 32, and the preliminary tool was designed [17].

A REDCap® data base was generated for the multicentre data collection enabling the sending of online questionnaires and data entry in case report form by study midwives [17]. To minimise missing data relevant questions were mandatory. A self-reported antenatal online questionnaire including pregnancy history was developed, the validated German versions of the Childbirth Self-Efficacy Inventory (CBSEI, 15 labour items published by Schmidt et al. 2015) [26,27] and the

Cambridge-Worry Scale (CWS) [28,29] were used for the preliminary validation of the tool. The responses to the preliminary tool, socio-demographic data as well as labour and birth data were entered by the study midwives in the case report form. Finally, a self-reported postnatal online questionnaire was sent to the participants six weeks after birth. All instruments were tested in a pilot phase with n = 73 participants.

## Recruitment and data collection

Potential participants were recruited during the second half of pregnancy by the study midwives on-site at the study centres. Eligible women were approached after booking the hospital, during antenatal check-ups, antenatal classes, and acupuncture appointments. The preliminary tool, consisting of 32 items for structured assessment, was applied during the first contact with the parturient in early labour, whether this occurred via telephone or in person. Midwives administered the questionnaire by posing the questions verbally. If women called repeatedly, the tool could be utilised multiple times.

## Ethics

Participants were informed verbally and in writing about the purpose of the study and their right to withdraw at any time. They gave written consent to the participation in the study. The Ethics Committees of Zurich as well as North-western and Central Switzerland classified the study as a clinical trial and approved it in July 2021 (BASEC-Nr. 2021–00687).

## Statistical analyses

Descriptive analysis was done computing absolute and relative frequencies for categorical variables and median as well as minimum/maximum for metric variables due to their non-normal distribution. The comparison between characteristics of cases with completed GebStart-tool and dropouts was performed using chi squared tests for categorical and Mann-Withey-U-tests for metric characteristics according to the type and distribution of variables.

Histograms were used to show the distribution of response categories and the number of missing answers per item and thereby, to explore the usefulness of each item to assess differences between participants. The instrument became more complex than originally intended and its character was ultimately clearly formative in nature, meaning that constructs or domains were the results of a group of items [30]. Factor analysis was therefore not appropriate for item reduction. Nevertheless, a scree plot was created to explore the number of factors which could advise the literature-based categorisation of the items into domains (S1 Fig). This subdivision resulted in the domains 'Physical symptoms', 'Emotional state', 'Self-management' and 'Resources' which were used for further analyses. In a first step, Partial Least Square Equation (PLS-SEM) modelling was explored but led to little information to constrain the selection of items (S2 Table).

Subsequently, we used Cox regression modelling with the dependent variables which described care needs (duration between the completion of the GebStart-tool and hospital admission, onset of active labour, first medical pain management and first alternative pain management). For women with onset of active labour before the completion of the GebStart-tool, the negative time intervals were replaced with a value of 0.001 hrs to maximise the sample included in the models because Cox regression modelling excludes cases with negative time intervals. Cox regression modelling was performed for each of the domains 'Physical symptoms', 'Emotional state', 'Self-management' and 'Resources' and was adjusted for age. Stepwise forward selection adding the items of the tool as whole categorical variables and using a threshold of p > 0.2 was applied. As the statistical programme kept single response categories in the models, final models were computed with the previously selected variables without forward selection and have also been adjusted for age, which was significantly associated with several outcomes. This enabled the retention of all response categories of the items. Hazard ratios > 1 indicate shorter time span and were interpreted as earlier care needs. The relevance of the items was estimated by looking at the number of significant associations with the outcome variables having a Hazard ratio >1. Ratings between very low and very high were assigned.

Moreover, multinominal logistic regressions models with the outcome care decision 'stay at home – keep in contact – hospital admission' for each domain ('Physical symptoms', 'Emotional state', 'Self-management' and 'Resources') were used to investigate the predictive value of the corresponding items for this outcome. Models were adjusted for age, and 'keep in contact' was determined as the reference category because it had the largest subsample. Risk ratios were reported as proposed for multinominal regression [31]. A synthesis of the results to Cox regression and multinominal regression models was performed to reduce items. For items with heterogenous results in this synthesis, the relevance for clinical practice, which was assessed for face validity in the development phase of the tool, was used to select the better one from similar items. These decisions were also driven by the wish to keep a comprehensive tool. Considerations for item reduction were narratively described.

Because of sample size restrictions and minor item adaptions for the final tool, a preliminary validation was envisaged. For this, Cox regression models again with the dependent variables which describe care needs (duration between the completion of the GebStart-tool and hospital admission, onset of active labour, first medical pain management and first alternative pain management) as well as a multinominal regression with the outcome care decision were used. The total score with the sum of all items considered for the reduced version of the GebStart-tool was included as a predictor and the models were adjusted for sociodemographic variables. Additionally, we used Generalised estimating equation (GEE) models to assess temporal trajectories of the total score of the GebStart-tool. For assessing convergent validity between items related to the emotional state and the 15 labour items of the CBSEI published by Schmidt et al. [26,27] as well as the CWS [28,29], Spearman's rho correlations were applied. As some women were admitted to the hospital in active labour, sensitivity analysis was performed for the Cox regression models, the multinominal regressions, as well as the Generalised estimating equation (GEE) models used for the preliminary validation excluding women with cervical dilatation >6 cm at the first vaginal examination after hospital admission. The threshold of 6 cm was decided because the World Health Organisation (WHO) defined the end of early labour at 5 cm [4] and the cervical dilatation could not be assessed at the time of the completion of the tool. The median time interval between the first completion of the tool and hospital admission lasted 3.5 hours.

Cut off points were calculated using multinomial regression modelling and plotting the total score against estimated probabilities as proposed by Bersabé and Rivas [32]. All analyses were performed using Stata 17 (StataCorp, College Town, USA) and p-values <0.05 were considered to be statistically significant.

## Results

A total of n = 627 pregnant women were recruited during the main study phase of whom n = 394 (62.8%) had at least one completed GebStart-tool. Two tools were applied for n = 172 participants, three for n = 43, four for n = 11 and five for n = 1 participant(s). Since women were recruited during prengnancy a high number of women had to be excluded by the time they gave birth (n = 233, 37.2%) due to either medical induction, caesarean section before onset of labour or in some women, because, the GebStart-tool was not completed, even though they had a spontaneous onset of labour. Reasons for the latter cases (n = 95, 15.2%) were that the workload in the study sites was high, hospital admission was urgent or completing the tool was forgotten.

### Characteristics of participants

Participants with completed GebStart-tool included in these analyses were in a median 33.0 years old and 74.1% of them were Swiss (Table 1). Nearly all (98.7%) were living with their partner, husband or wife and 60.6% had a university degree. Moreover, a great majority of women were employed (96.8%) with a median workload of 100%. Most families (85.2%) managed very well or well with their annual family income. Furthermore, more than three quarters of participants (79.4%) had public health insurance. Regarding sociodemographic characteristics, participants without completed GebStart-tool who dropped out of the study differed only in the fact that they were less often Swiss compared to those with completed tool (63.9% vs 74.1%, p = 0.009, Table 1).

**Table 1. Characteristics of participants with applied tools in comparison to dropouts.**

| Characteristics | Participants with tool applied n = 394 | Exclusion without tool applied n = 233 | p-value |
|---|---|---|---|
| *Sociodemographic characteristics* | | | |
| **Age** in years, md (min – max) | 33.0 (20.0-47.0) | 33.0 (21.0-44.0) | 0.509[1] |
| **Nationality*** | | | |
| Swiss, n (%) | 280 (74.1) | 138 (63.9) | **0.009**[2] |
| **Way of life***, n (%) | | | |
| Living with partner/husband/wife | 373 (98.7) | 214 (99.1) | 0.361[2] |
| Single | 2 (0.5) | 2 (0.9) | |
| Other | 3 (0.8) | 0 | |
| **Highest education level***, n (%) | | | |
| University | 229 (60.6) | 122 (56.5) | 0.328[2] |
| **Employed*** | | | |
| Yes, n (%) | 366 (96.8) | 211 (97.7) | 0.545[2] |
| **Workload**[3] in %, md (min – max) | 100 (10-105) | 100 (40-100) | 0.465[1] |
| **Management with the annual family income***, n (%) | | | |
| Very good/good | 322 (85.2) | 188 (87.0) | 0.170[2] |
| Moderate/ difficult | 49 (13.0) | 20 (9.3) | |
| Did not want to comment it | 7 (1.9) | 8 (3.7) | |
| **Health insurance** | | | |
| Public insurance | 312 (79.4) | 167 (73.9) | 0.128[2] |
| Semiprivate insurance | 63 (16.0) | 38 (16.8) | |
| Private insurance | 17 (4.3) | 20 (8.9) | |
| Other | 1 (0.3) | 1 (0.4) | |
| *Perinatal characteristics* | | | |
| **Gravidity**, n (%) | | | |
| Primigravidae | 325 (82.7) | 199 (87.3) | 0.406[2] |
| **Gestational weeks at recruitment,** md (min – max) | 36.0 (26.0-40.9) | 35.9 (29.0-40.0) | 0.132[1] |
| **Gestational weeks at birth,** md (min – max) | 40.0 (35.9-42.1) | 40.3 (35.0-42) | 0.065[1] |
| **Pregnancy planned**, n (%) | | | |
| Yes | 321 (84.9) | 186 (86.1) | 0.718[2] |
| Not at the present time, but not undesirable | 49 (13.0) | 24 (11.1) | |
| No | 8 (2.1) | 6 (2.8) | |
| **Antenatal care provider***, n (%) | | | |
| Gynaecologist | 319 (84.4) | 178 (82.4) | 0.949[2] |
| Midwife | 1 (0.3) | 1 (0.5) | |
| Gynaecologist and midwife | 56 (14.8) | 35 (16.2) | |
| Physician | 1 (0.3) | 1 (0.5) | |
| Other | 1 (0.3) | 1 (0.5) | |
| **Antenatal preparation*** | | | |
| Yes, n (%) | 384 (97.5) | 231 (99.1) | 0.138 |
| **Onset of early labour**, n (%) | | | |
| Spontaneous | 348 (89.0) | 76 (33.9) | **< 0.001**[2] |
| Natural or mechanical stimulation | 33 (8.4) | 19 (8.5) | |
| Medical induction | 10 (2.6) | 116 (51.8) | |
| Caesarean section before onset of labour | 0 | 13 (5.8) | |

*(Continued)*

**Table 1.** (Continued)

| Characteristics | Participants with tool applied n = 394 | Exclusion without tool applied n = 233 | p-value |
|---|---|---|---|
| **Rupture of membranes**, n (%) | | | |
| Premature rupture | 150 (38.4) | 70 (31.8) | **0.043**[2] |
| Spontaneous, during active labour | 179 (45.8) | 98 (44.6) | |
| Amniotomy | 62 (15.9) | 52 (23.6) | |
| **Labour augmentation with oxytocin during labour**[4], n (%) | 205 (52.3) | 123 (60.0) | 0.078[2] |
| **Opioid administration**[4], n (%) | 132 (33.6) | 86 (39.5) | 0.147 |
| **Epidural analgesia**[4], n (%) | 199 (50.9) | 139 (67.2) | **< 0.001**[2] |
| **Mode of birth**, n (%) | | | |
| Spontaneous vaginal | 239 (60.8) | 98 (43.4) | **< 0.001**[2] |
| Instrumental vaginal (ventouse) | 92 (23.4) | 59 (26.1) | |
| Caesarean section during labour | 61 (15.5) | 54 (23.9) | |
| Caesarean section before onset of labour | 1 (0.3) | 15 (6.6) | |
| **Birth weight** in gram, md (min – max) | 3310 (2340-4740) | 3370 (2070-5110) | 0.355[1] |
| *Scores of validation scales* | | | |
| **CBSEI-Score**[5], md (min – max) | 7.1 (3.7-9.9) | 7.1 (2.9-10.0) | 0.675 |
| **CWS-Score**[6], md (min – max) | 26.0 (2.0-73.0) | 27.0 (2.0-63.0) | 0.622 |

\* 5.3% missing because of missing antenatal questionnaire.

[1] Mann Whitney U-Test.

[2] Chi Quadrat Test.

[3] A 100% workload corresponds to full-time employment and lower percentages to a part time position.

[4] Caesarean section before onset of labour excluded from these analyses.

[5] Childbirth Self-Efficacy Inventory, 15 questions of the labour subscale about how helpful women consider the behaviour.

[6] Cambridge Worry Scale.

In accordance with the inclusion criteria, all participants were nulliparous and 82.7% with completed tools were primigravidae (Table 1). The median gestational week at recruitment was 36.0 and at birth 40.0. In n = 2 participants with late preterm birth, the GebStart-tool was completed although this was one of the exclusion criteria. As data were already collected, these cases were not excluded from the analysis. Most included participants had a planned pregnancy (84.9%). Furthermore, the primary antenatal care provider was the gynaecologist for a large majority of women (84.4%) and most of them attended an antenatal birth preparation course (97.5%). In most participants (89.0%), onset of labour was spontaneous. For n = 10 women with unplanned medical inductions at the time of recruitment (2.6%), the GebStart-tool was completed with signs of onset of labour before the induction. Almost half of the participants experienced spontaneous ruptures of membranes during labour (45.8%) and in slightly more than half (52.3%), labour was augmented with synthetic oxytocin. For pain management, one third of women received opioids (33.6%) and more than half had an epidural analgesia (50.9%). While most participants had a spontaneous vaginal birth (60.8%), 23.4% experienced an instrumental birth and 15.5% an unplanned caesarean section during labour. For one woman with caesarean section before the onset of active labour (0.3%), the GebStart-tool was completed because of signs of onset of labour. The median infant birth weight was 3,310 grams. With regard to perinatal characteristics, women who dropped out of the study had a spontaneous onset of labour significantly less often (p < 0.001), more often an amniotomy (p = 0.043) or an epidural analgesia (p < 0.001), but less often a spontaneous birth (p < 0.001, Table 1). All the other characteristics did not differ between participants with the applied tool and dropouts.

Regarding the validation items which were completed during pregnancy, the median CBSEI sub score (15 items) was 7.1 points and the median CWS score 26.0 points.

## Item reduction

The explorative analysis for the distribution of response options showed almost no variation for item 13 (Fig 1). The response variations for item 29 and item 30, both about the support received during early labour at home, was low, but slightly higher for item 29. Additionally, item 25 was not completed in 83 cases, which indicated that this item was unsuitable for the GebStart-tool. Therefore, the items 13, 25 and 30 were excluded at this point and were not taken into consideration for further analyses.

The scree plot for factor analysis indicated four factors (S1 Fig), which supported the theoretical subdivision of items into four domains. The Cox regression models for each of the domains 'Physical symptoms', 'Emotional state', 'Self-management' and 'Resources' and the duration between the completion of the GebStart-tool and events during the early labour process (timing of hospital admission, onset of active labour, first medical and first alternative pain management) showed variations in the numbers of significant associations with HR > 1 (Table 2). Some items did not have any associations with one of the outcomes (Item 3, 14, 16, 21); and others showed associations with all the outcomes (Items

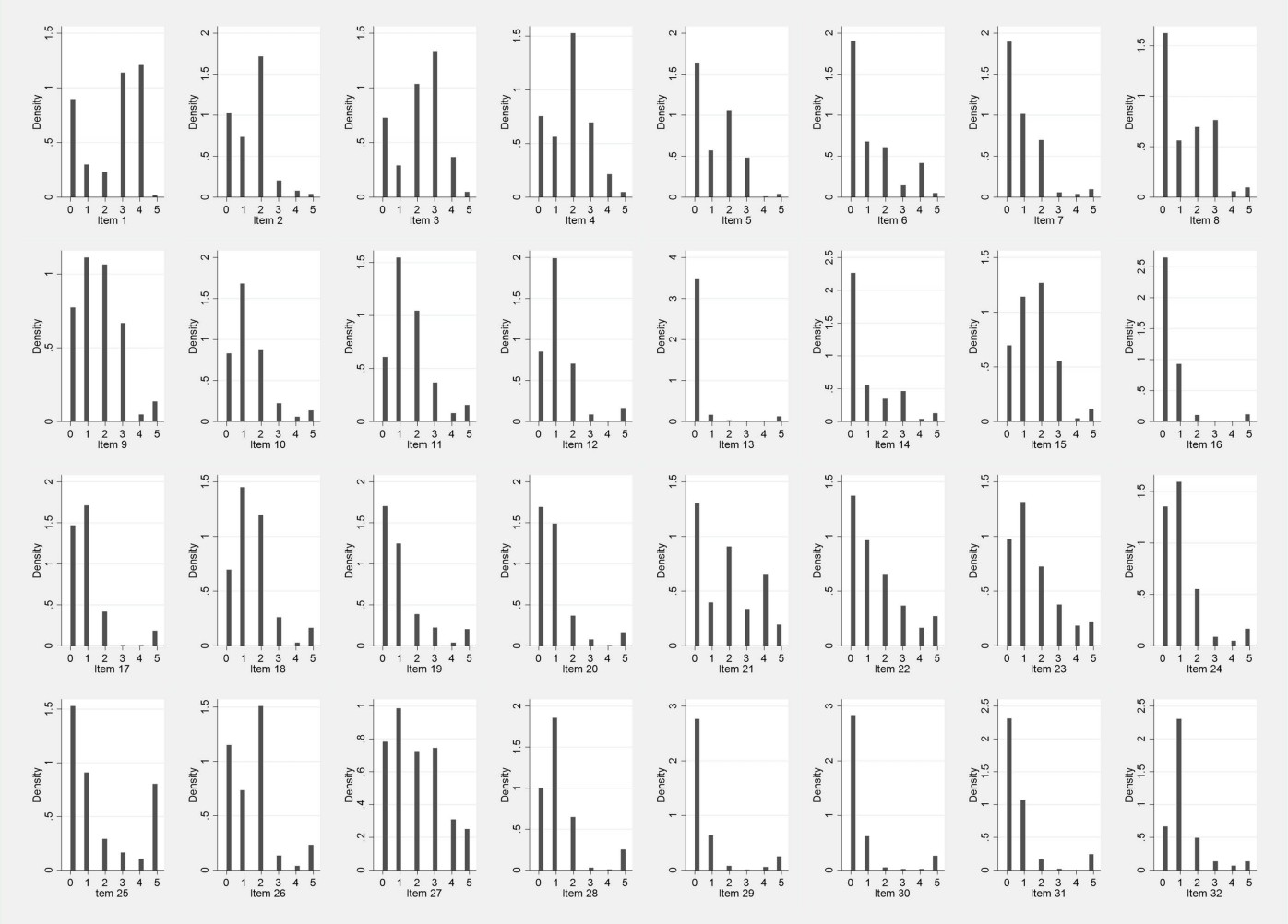

**Fig 1. Distributions of response categories and missing responses in Item 1-Item32 of the first completed tool.**

**Table 2. Time intervals between completing the GebStart-tool and hospital admission, onset of active labour, medical pain management and alternative pain management.**

| Items | Aspect, de-scription | Duration tool - hospital admission HR [95% CI] | Duration tool – onset of active labour HR [95% CI] | Duration tool – first medical pain management HR [95% CI] | Duration tool – first alternative pain management HR [95% CI] | Relevance |
|---|---|---|---|---|---|---|
| **Physical symptoms** | | | | | | |
| Age | | 1.04 [1.01, 1.08] * | 1.03 [1.00, 1.07] | 1.07 [1.03, 1.11] *** | 1.05 [1.01, 1.09] * | |
| Item 1 | Contractions | 0 (ref.) | 0: (ref.) | 0: (ref.) | 0: (ref.) | Very high |
| | | 1: 1.15 [0.58, 2.28] | 1: 0.84 [0.47, 1.53] | 1: 0.81 [0.41, 1.61] | 1: 1.22 [0.60, 2.49] | |
| | | 2: 1.96 [0.88, 4.34] | 2: 0.97 [0.47, 2.02] | 2: 1.18 [0.54, 2.56] | 2: 2.04 [0.87, 4.77] | |
| | | 3: 2.05 [1.05, 4.03] * | 3: 1.42 [0.74, 2.74] | 3: 1.70 [0.87, 3.30] | 3: 2.96 [1.36, 6.43] ** | |
| | | 4: 2.66 [1.34, 5.31] ** | 4: 1.67 [0.85, 3.28] | 4: 2.00 [1.02, 3.94] * | 4: 3.30 [1.49, 7.34] ** | |
| Item 2 | Contractions | | 0: (ref.) | | 0: (ref.) | Medium |
| | | | 1: 1.44 [0.86, 2.42] | | 1: 0.95 [0.54, 1.68] | |
| | | | 2: 1.54 [0.92, 2.59] | | 2: 1.21 [0.69, 2.14] | |
| | | | 3: 1.23 [0.63, 2.39] | | 3: 0.77 [0.37, 1.59] | |
| | | | 4: 0.42 [0.15, 1.12] | | 4: 0.23 [0.07, 0.77] * | |
| Item 3 | Contractions | | | | | Very low |
| Item 4 | Contractions | 0: (ref.) | 0: (ref.) | 0: (ref.) | 0: (ref.) | Very high |
| | | 1: 1.11 [0.61, 2.02] | 1: 1.31 [0.76, 2.26] | 1: 2.11 [1.14, 3.91] * | 1: 1.40 [0.71, 2.73] | |
| | | 2: 1.08 [0.53, 2.21] | 2: 1.46 [0.78, 2.72] | 2: 2.12 [1.03, 4.35] * | 2: 0.98 [0.44, 2.16] | |
| | | 3: 1.52 [0.71, 3.25] | 3: 1.79 [0.92, 3.51] | 3: 3.23 [1.48, 7.07] ** | 3: 1.36 [0.59, 3.15] | |
| | | 4: 2.49 [1.04, 5.97] * | 4: 2.93 [1.28, 6.87] * | 4: 6.53 [2.43, 17.57] *** | 4: 2.92 [1.08, 7.90] * | |
| Item 5 | Contractions | 0: (ref.) | | 0: (ref.) | 0: (ref.) | High |
| | | 1: 1.15 [0.79, 1.67] | | 1: 1.10 [0.73, 1.64] | 1: 0.88 [0.57, 1.36] | |
| | | 2: 1.37 [0.96, 1.96] | | 2: 1.41 [0.97, 2.06] | 2: 1.43 [0.95, 2.14] | |
| | | 3: 1.89 [1.20, 2.98] ** | | 3: 1.43 [0.83, 2.46] | 3: 1.67 [1.01, 2.74] * | |
| | | 4: - | | 4: - | 4: - | |
| Item 6 | Vaginal discharge | 0: (ref.) | 0: (ref.) | | 0: (ref.) | High |
| | | 1: 1.11 [0.79, 1.54] | 1: 1.03 [0.74, 1.43] | | 1: 1.27 [0.87, 1.84] | |
| | | 2: 1.60 [1.08, 2.38] * | 2: 1.42 [0.94, 2.15] | | 2: 2.00 [1.27, 3.15] ** | |
| | | 3: 1.01 [0.49, 2.08] | 3: 1.47 [0.74, 2.91] | | 3: 1.50 [0.68, 3.31] | |
| | | 4: 1.53 [0.98, 2.40] | 4: 1.78 [1.11, 2.85] * | | 4: 1.92 [1.13, 3.24] | |
| Item 7 | Vaginal discharge | | | 0: (ref.) | | Low |
| | | | | 1: 1.60 [1.05, 2.43] * | | |
| | | | | 2: 1.36 [0.96, 1.94] | | |
| | | | | 3: 2.89 [0.91, 9.22] | | |
| | | | | 4: 1.94 [0.68, 5.57] | | |
| Item 8 | Vaginal discharge | 0: (ref.) | 0: (ref.) | 0: (ref.) | 0: (ref.) | Very high |
| | | 1: 1.52 [1.08, 2.12] * | 1: 1.30 [0.92, 1.83] | 1: 1.60 [1.10, 2.32] * | 1: 1.68 [1.16, 2.45] ** | |
| | | 2: 1.21 [0.87, 1.68] | 2: 1.23 [0.89, 1.72] | 2: 1.19 [0.84, 1.70] | 2: 1.36 [0.94, 1.98] | |
| | | 3: 2.02 [1.40, 2.93] *** | 3: 1.32 [0.92, 1.91] | 3: 1.37 [0.92, 2.06] | 3: 1.94 [1.26, 2.97] ** | |
| | | 4: 1.59 [0.42, 6.05] | 4: 3.28 [0.94, 11.40] | 4: 3.28 [0.75, 14.40] | 4: 0.71 [0.15, 3.32] | |

*(Continued)*

| Items | Aspect, de-scription | Duration tool - hospital admission HR [95% CI] | Duration tool – onset of active labour HR [95% CI] | Duration tool – first medical pain management HR [95% CI] | Duration tool – first alternative pain management HR [95% CI] | Relevance |
|---|---|---|---|---|---|---|
| Item 9 | Sleep | 0: (ref.) | 0: (ref.) | 0: (ref.) | 0: (ref.) | Very high |
| | | 1: 0.55 [0.38, 0.78] ** | 1: 0.47 [0.33, 0.68] *** | 1: 0.58 [0.39, 0.87] ** | 1: 0.48 [0.32, 0.72] *** | |
| | | 2: 0.87 [0.60, 1.25] | 2: 0.88 [0.60, 1.27] | 2: 1.06 [0.71, 1.58] | 2: 0.95 [0.63, 1.44] | |
| | | 3: 0.69 [0.46, 1.03] | 3: 0.76 [0.50, 1.16] | 3: 0.79 [0.50, 1.25] | 3: 0.69 [0.44, 1.11] | |
| | | 4: 3.05 [1.03, 9.02] * | 4: 3.64 [1.08, 12.24] * | 4: 1.94 [0.65, 5.81] | 4: 28.82 [4.08, 203.62] ** | |
| Item 10 | Fitness | | 0 (ref.) | | 0: (ref.) | Medium |
| | | | 1: 1.17 [0.81, 1.67] | | 1: 1.03 [0.73, 1.46] | |
| | | | 2: 1.09 [0.68, 1.76] | | 2: 1.34 [0.89, 2.04] | |
| | | | 3: 0.73 [0.34, 1.56] | | 3: 0.58 [0.29, 1.13] | |
| | | | 4: 0.12 [0.02, 0.72] * | | 4: 1.05 [0.34, 3.26] | |
| Item 11 | Fitness | | 0: (ref.) | | | Low |
| | | | 1: 1.01 [0.67, 1.50] | | | |
| | | | 2: 1.23 [0.76, 1.98] | | | |
| | | | 3: 0.69 [0.36, 1.33] | | | |
| | | | 4: 7.34 [1.58, 34.21] * | | | |
| Item 12 | Gastro-intestinal | | 0: (ref.) | | 0: (ref.) | Medium |
| | | | 1: 1.07 [0.80, 1.43] | | 1: 0.93 [0.67, 1.30] | |
| | | | 2: 1.38 [0.96, 1.99] | | 2: 1.03 [0.69, 1.54] | |
| | | | 3: 2.06 [0.91, 4.64] | | 3: 2.63 [1.09, 6.31] | |
| | | | 4: - | | 4: - | |
| Item 14 | Gastro-intestinal | | | 0: (ref.) | | Very low |
| | | | | 1: 0.88 [0.62, 1.24] | | |
| | | | | 2: 0.51 [0.32, 0.81] ** | | |
| | | | | 3: 0.66 [0.43, 1.00] * | | |
| | | | | 4: 0.54 [0.18, 1.60] | | |
| Item 15 | Foetal movement | | | 0: (ref.) | | Low |
| | | | | 1: 1.41 [0.99, 2.01] | | |
| | | | | 2: 1.23 [0.86, 1.75] | | |
| | | | | 3: 0.88 [0.57, 1.37] | | |
| | | | | 4: 2.68 [0.34, 21.34] | | |
| Item 16 | Foetal movement | 0: (ref.) | | | 0: (ref.) | Low |
| | | 1: 1.16 [0.90, 1.51] | | | 1: 1.20 [0.91, 1.59] | |
| | | 2: 0.45 [0.22, 0.91] * | | | 2: 0.44 [0.20, 0.96] * | |
| | | 3: - | | | 3: - | |
| | | 4: - | | | 4: - | |

*(Continued)*

| Items | Aspect, de-scription | Duration tool - hospital admission HR [95% CI] | Duration tool – onset of active labour HR [95% CI] | Duration tool – first medical pain management HR [95% CI] | Duration tool – first alternative pain management HR [95% CI] | Relevance |
|---|---|---|---|---|---|---|
| **Emotional symptoms** | | | | | | |
| Age | | 1.02 [0.99, 1.05] | 1.02 [0.99, 1.05] | 1.03 [1.00, 1.07] * | 1.02 [0.99, 1.05] | |
| Item 17 | Confidence | 0: (ref.) | 0: (ref.) | 0: (ref.) | 0: (ref.) | Very high |
| | | 1: 0.88 [0.68, 1.12] | 1: 0.87 [0.68, 1.12] | 1: 0.98 [0.76, 1.28] | 1: 0.93 [0.72, 1.22] | |
| | | | 2: 0.48 [0.31, 0.74] ** | 2: 0.53 [0.34, 0.84] ** | 2: 0.48 [0.30, 0.77] ** | 2: 0.39 [0.23, 0.65] *** |
| | | | 3: 11.66 [1.37, 99.18] * | 3: 0.18 [0.02, 1.97] | 3: 2.62 [0.26, 26.12] | 3: 0.24 [0.03, 2.08] |
| | | | 4: 9.50 [1.15, 78.63] * | 4: 1.30 [0.16, 10.77] | 4: 0.68 [0.09, 5.17] | 4: 1.04 [0.13, 8.53] |
| Item 18 | Overall emotional state | 0: (ref.) | 0: (ref.) | | 0: (ref.) | Medium |
| | | 1: 1.01 [0.72, 1.41] | 1: 1.12 [0.80, 1.58] | | 1: 1.12 [0.78, 1.61] | |
| | | 2: 0.99 [0.69, 1.42] | 2: 1.50 [1.04, 2.15] * | | 2: 1.19 [0.80, 1.76] | |
| | | 3: 1.06 [0.59, 1.90] | 3: 0.88 [0.45, 1.73] | | 3: 1.38 [0.72, 2.66] | |
| | | 4: 0.04 [0.00, 0.46] ** | 4: - | | 4: 0.02 [0.00, 0.23] ** | |
| Item 19 | Safe at home | 0: - | 0: (ref.) | | 0: (ref.) | High |
| | | 1: 1.10 [0.84, 1.44] | 1: 0.86 [0.65, 1.14] | | 1: 0.91 [0.67, 1.22] | |
| | | 2: 1.66 [1.10, 2.52] * | 2: 1.90 [1.20, 3.01] ** | | 2: 1.13 [0.71, 1.81] | |
| | | 3: 1.01 [0.58, 1.78] | 3: 0.66 [0.35, 1.26] | | 3: 1.65 [0.84, 3.24] | |
| | | 4: 1.71 [0.44, 6.61] | 4: 0.53 [0.13, 2.11] | | 4: 9.42 [2.27, 39.15] ** | |
| Item 20 | Worrying | 0: - | 0: (ref.) | 0: (ref.) | | Medium |
| | | 1: | 1: 0.87 [0.67, 1.12] | 1: 1.07 [0.82, 1.39] | | |
| | | 2: | 2: 0.76 [0.50, 1.17] | 2: 0.64 [0.40, 1.02] | | |
| | | 3: | 3: 2.96 [0.85, 10.32] | 3: 3.33 [0.99, 11.17] | | |
| | | 4: | 4: - | 4: - | | |
| Item 22 | Feeling at home | 0: (ref.) | 0: (ref.) | 0: (ref.) | 0: (ref.) | Very high |
| | | 1: 1.40 [1.05, 1.87] * 2: 1.96 [1.39, 2.77] *** | 1: 1.44 [1.06, 1.94] * 2: 1.94 [1.35, 2.78] *** | 1:1.37 [1.01, 1.86] * 2: 2.65 [1.80, 3.91] *** | 1: 1.32 [0.96, 1.81] 2: 1.82 [1.24, 2.67] ** | |
| | | 3: 2.06 [1.35, 3.16] ** | 3: 2.25 [1.47, 3.43] *** | 3: 2.71 [1.69, 4.35] *** | 3: 2.07 [1.29, 3.34] ** | |
| | | 4: 7.45 [3.63, 15.28] *** | 4: 6.76 [3.10, 14.76] *** | 4: 7.58 [4.06, 14.16] *** | 4: 3.65 [1.67, 8.03] ** | |
| **Self-management** | | | | | | |
| Age | | 1.02 [0.99, 1.05] | 1.02 [0.99, 1.05] | 1.04 [1.00, 1.07] * | 1.02 [0.99, 1.05] | |
| Item 21 | Reason for call | | | | | Very low |
| Item 23 | Distraction | 0: (ref.) | 0: (ref.) | 0: (ref.) | 0: (ref.) | Very high |
| | | 1: 1.02 [0.76. 1.37] | 1: 1.17 [0.90, 1.53] | 1: 1.18 [0.86, 1.63] | 1: 1.01 [0.74, 1.37] | |
| | | 2: 1.48 [1.02, 2.15] * | 2: 1.70 [1.25, 2.32] ** | 2: 2.12 [1.43, 3.15] *** | 2: 1.62 [1.10, 2.37] * | |
| | | 3: 1.37 [0.88, 2.13] | 3: 1.40 [0.95, 2.06] | 3: 1.72 [1.08, 2.73] * | 3: 1.26 [0.81, 1.98] | |
| | | 4: 1.10 [0.57, 2.14] | 4: 4.48 [2.65, 7.59] *** | 4: 1.79 [0.77, 4.12] | 4: 1.30 [0.66, 2.57] | |

*(Continued)*

| Items | Aspect, de-scription | Duration tool - hospital admission HR [95% CI] | Duration tool – onset of active labour HR [95% CI] | Duration tool – first medical pain management HR [95% CI] | Duration tool – first alternative pain management HR [95% CI] | Relevance |
|---|---|---|---|---|---|---|
| Item 24 | Handling | 0: (ref.) | | 0: (ref.) | 0: (ref.) | High |
| | | 1: 1.16 [0.88, 1.53] | | 1: 1.22 [0.91, 1.64] | 1: 1.42 [1.07, 1.88] * | |
| | | 2: 1.37 [0.95, 1.96] | | 2: 1.54 [1.03, 2.30] * | 2: 1.44 [0.97, 2.13] | |
| | | 3: 5.85 [2.65, 12.91] *** | | 3: 5.59 [2.15, 14.57] *** | 3: 2.61 [1.14, 5.99] * | |
| | | 4: 2.35 [0.81, 6.77] | | 4: 1.79 [0.57, 5.65] | 4: 2.96 [0.98, 8.93] | |
| Item 27 | Preferences management | 0: (ref.) | | | | low |
| | | 1: 0.88 [0.65, 1.20] | | | | |
| | | 2: 1.17 [0.82, 1.66] | | | | |
| | | 3: 1.16 [0.82, 1.65] | | | | |
| | | 4: 1.75 [1.06, 2.89] * | | | | |
| **Resources** | | | | | | |
| Age | | 1.02 [0.98, 1.05] | 1.02 [0.99, 1.05] | 1.02 [0.99, 1.06] | 1.02 [0.98, 1.05] | |
| Item 26 | Attitude | 0: (ref.) | | 0: (ref.) | | Medium |
| | | 1: 1.32 [0.96, 1.81] | | 1: 1.29 [0.90, 1.86] | | |
| | | 2: 1.18 [0.90, 1.54] | | 2: 1.50 [1.10, 2.05] * | | |
| | | 3: 1.49 [0.79, 2.82] | | 3: 2.03 [1.00, 4.14] | | |
| | | 4: 7.65 [2.60, 22.52] *** | | 4: 4.29 [1.06, 17.32] | | |
| Item 28 | Preparation | 0: (ref.) | 0: (ref.) | 0: (ref.) | 0: (ref.) | Very high |
| | | 1: 0.90 [0.69, 1.19] | 1: 1.00 [0.78, 1.28] | 1: 0.96 [0.71, 1.30] | 1: 1.05 [0.80, 1.38] | |
| | | 2: 0.69 [0.49, 0.98] * | 2: 0.69 [0.49, 0.96] * | 2: 0.63 [0.44, 0.92] * | 2: 0.67 [0.46, 0.96] * | |
| | | 3: 0.75 [0.16, 3.41] | 3: 1.87 [0.45, 7.76] | 3: 0.59 [0.09, 4.01] | 3: 1.55 [0.48, 5.02] | |
| | | 4: 12.25 [0.71, 210.20] | 4: 1.34 [0.08, 22.09] | 4: 1.91 [0.11, 32.95] | 4: 0.82 [0.11, 5.91] | |
| Item 29 | Support | 0: (ref.) | | 0: (ref.) | | Medium |
| | | 1: 0.86 [0.63, 1.18] | | 1: 0.86 [0.61, 1.22] | | |
| | | 2: 2.94 [1.29, 6.71] * | | 2: 3.72 [1.37, 10.07] * | | |
| | | 3: 0.22 [0.03, 1.68] | | 3: 0.25 [0.03, 1.96] | | |
| | | 4: 2.20 [0.88, 5.48] | | 4: 2.13 [0.86, 5.32] | | |
| Item 31 | Companion | 0: (ref.) | 0: (ref.) | 0: (ref.) | 0: (ref.) | Very high |
| | | 1: 1.35 [1.04, 1.75] * | 1: 1.29 [1.02, 1.63] * | 1: 1.48 [1.10, 1.98] ** | 1: 1.36 [1.04, 1.77] * | |
| | | 2: 1.30 [0.77, 2.19] | 2: 1.26 [0.75, 2.10] | 2: 1.22 [0.69, 2.18] | 2: 1.91 [1.13, 3.22] * | |
| | | 3: 0.53 [0.07, 3.90] | 3: 0.51 [0.07, 3.72] | 3: 0.58 [0.08, 4.32] | 3: omitted | |
| | | 4: - | 4: - | 4: - | 4: - | |
| Item 32 | Distance to care facility | 0: (ref.) | | 0: (ref.) | | Medium |
| | | 1: 1.10 [0.82, 1.48] | | 1: 1.14 [0.82, 1.59] | | |
| | | 2: 1.35 [0.91, 1.20] | | 2: 1.26 [0.81, 1.96] | | |
| | | 3: 2.31 [1.10, 4.83] * | | 3: 4.52 [2.02, 10.12] *** | | |
| | | 4: 2.09 [0.82, 5.30] | | 4: 1.48 [0.57, 3.84] | | |

* p < 0.05.

** p < 0.01.

*** p < 0.001.

1, 4, 8, 9, 17, 22, 23). The classification of the relevance was based on the number of significant associations of items with the describing care needs and having a HR > 1.

In the multinomial regression models investigating the associations of items with the care decision, the items 1, 5, 7, 8, 10, 12 and 14 remained in the final multinominal regression model for 'Physical symptoms', and item 22 in the model for 'Emotional symptoms' (Table 3). The final model for 'Self-management' contained the items 21, 24 as well as 27; and the items 29, 31 and 32 were kept in the model for "Resources".

A synthesis of the above-described different statistical analyses and the face validity assessed during the development phase showed a clear picture to include or exclude some items in the reduced instrument (Table 4). However, for others, results were heterogenous. Additional justifications to reduce items were added in Table 4 to ensure that as few aspects as possible were lost and that for similar items, the one with the best performance in the analyses and/or face validity was chosen.

The reduced GebStart-tool consists of 15 items (Table 5). Of the 394 first applied tools, 346 were complete (87.8%). The median total score of these complete first tools was 16 points with a minimum of 3 and a maximum of 34 points out of the maximum possible of 60 points.

### Cutt-off points for the reduced instrument

Using multinomial regression modelling with the remaining 15 items, cut-off points of 22 for the decision 'Keep in contact and 33 for the decision 'Hospital admission' were computed. These cut-off points were also shown in the plot of the total score against the probability to be in one of the management decision categories (Fig 2). However, for the second cut-off point, the plotted curves did not intersect, even though the values could be calculated according to the suggestion by Bersabé & Riva [32]. In order not to keeping women from receiving care and for safety reasons, the calculated second cut-off point at 33 was retained.

### Designing the final version of the GebStart-tool

The final version of the GebStart-tool was designed by using the remaining 15 items. Minor changes to two items were made by slightly rephrasing the question of item 7 and the answers of item 8, both about vaginal discharges. This was necessary, because in the preliminary tool, item 7 was formulated as a follow-up question to item 6. After eliminating item 6, its first partial question "Do you lose liquid from your vagina …" was added to item 7. This did not change the meaning of item 7 about the duration of the liquid loss. In the response options for question 8, "clear liquid" was added due to the feedback from the application in the study centres that this was missing. The final German version of the GebStart-tool (S1 File, S2 File) was designed to be completed by health professionals together with the parturients.

### Preliminary validation

The preliminary validation of the GebStart-tool was conducted with the 15 remaining items of the same data set. The total score of the reduced GebStart-tool for women who answered to all included items (n = 343) was significantly associated with the time intervals used to describe care needs: duration between completion of the tool and hospital admission (HR = 1.08, 95% CI [1.05–1.10], p < 0.001), onset of active labour (HR = 1.06, 95% CI [1.04–1.08], p < 0.001), first use of medical pain management (HR = 1.08, 95% CI [1.06–1.11], p < 0.001) and first use of alternative pain management (HR = 1.08, 95% CI [1.05–1.10], p < 0.001, S3 Table). Sensitivity analysis excluding women with cervical dilatation > 6 cm n = 270) showed similar, highly significant associations between the total score of the GebStart-tool and all intervals describing care needs (p < 0.001 for all associations). However, compared to the care decision 'Keep in contact', a higher total score of the reduced GebStart-tool was not significantly associated with a reduced risk for the decision 'Stay at home' (RR = 0.98, 95% CI [0.94–1.02], p = 421), but with a significantly higher risk for the decision 'Hospital admission' (RR = 1.13, 95% CI [1.05–1.22], p = 0.001) in the multinominal regression model (S4 Table). Again, sensitivity analysis excluding women with cervical dilatation > 6 cm showed similar results for the decision 'Hospital admission (RR = 1.17, 95% CI [1.08, 1.27, p < 0.001)

**Table 3. Item correlations with the care decision in multinominal logistic regression models.**

| Item | Aspect, description | Response options | Risk Ratio | 95% CI | p-value |
|---|---|---|---|---|---|
| **Physical symptoms** | | | | | |
| *Decision: 'Stay at home'* | | | | | |
| Age | | | 0.95 | 0.87-1.04 | 0.295 |
| Item 1 | Contractions | 1 | 3.01 | 0.74-12.23 | 0.124 |
| | | 2 | 34.24 | 6.13-191.10 | <0.001 |
| | | 3 | 17.74 | 4.93-63.85 | <0.001 |
| | | 4 | 12.94 | 3.43-48.85 | <0.001 |
| Item 5 | Contractions | 1 | 0.63 | 0.21-1.94 | 0.425 |
| | | 2 | 0.30 | 0.11-0.82 | 0.019 |
| | | 3 | 0.19 | 0.05-0.68 | 0.010 |
| | | 4 | – | – | – |
| Item 7 | Vaginal discharge | 1 | 0.16 | 0.06-0.40 | <0.001 |
| | | 2 | 0.13 | 0.06-0.31 | <0.001 |
| | | 3 | 0 | 0- | 0.991 |
| | | 4 | 0 | 0- | 0.999 |
| Item 8 | Vaginal discharge | 1 | 1.11 | 0.49-2.55 | 0.799 |
| | | 2 | 1.22 | 0.56-2.67 | 0.612 |
| | | 3 | 0.08 | 0.02-0.34 | 0.001 |
| | | 4 | 0 | 0- | 0.999 |
| Item 10 | Fitness | 1 | 1.16 | 0.50-2.68 | 0.729 |
| | | 2 | 0.68 | 0.27-1.77 | 0.433 |
| | | 3 | 0.31 | 0.08-1.20 | 0.091 |
| | | 4 | 0.20 | 0.01-6.93 | 0.372 |
| Item 12 | Gastro-intestinal | 1 | 0.52 | 0.23-1.17 | 0.113 |
| | | 2 | 0.44 | 0.17-1.15 | 0.095 |
| | | 3 | 1.98 | 0.24-16.51 | 0.527 |
| | | 4 | – | – | – |
| Item 14 | Gastro-intestinal | 1 | 1.51 | 0.60-3.77 | 0.381 |
| | | 2 | 0.56 | 0.19-1.69 | 0.303 |
| | | 3 | 0.44 | 0.18-1.10 | 0.079 |
| | | 4 | 0.26 | 0.02-2.90 | 0.271 |
| Constant | | | 6.54 | 0.27-160.70 | 0.251 |
| *Decision: 'Hospital admission'* | | | | | |
| Age | | | 1.04 | 0.92-1.18 | 0.512 |
| Item 1 | Contractions | 1 | 6.13 | 1.25-29.97 | 0.025 |
| | | 2 | 5.00 | 0.34-73.70 | 0.241 |
| | | 3 | 4.02 | 0.73-22.24 | 0.110 |
| | | 4 | 10.25 | 1.74-60.41 | 0.010 |
| Item 5 | Contractions | 1 | 0.16 | 0.02-1.26 | 0.082 |
| | | 2 | 0.42 | 0.10-1.80 | 0.241 |
| | | 3 | 0.78 | 0.15-4.04 | 0.770 |
| | | 4 | – | – | – |
| Item 7 | Vaginal discharge | 1 | 0.96 | 0.23-4.04 | 0.951 |
| | | 2 | 1.22 | 0.34-4.40 | 0.759 |
| | | 3 | 0 | 0- | 0.993 |
| | | 4 | 34.10 | 2.19-531.21 | 0.012 |

*(Continued)*

**Table 3.** (Continued)

| Item | Aspect, description | Response options | Risk Ratio | 95% CI | p-value |
|---|---|---|---|---|---|
| Item 8 | Vaginal discharge | 1 | 0.29 | 0.06-1.40 | 0.124 |
| | | 2 | 0.33 | 0.07-1.54 | 0.157 |
| | | 3 | 0.59 | 0.18-1.99 | 0.398 |
| | | 4 | 33.64 | 2.05-551.65 | 0.014 |
| Item 10 | Fitness | 1 | 2.20 | 0.50-9.59 | 0.296 |
| | | 2 | 3.38 | 0.72-15.86 | 0.122 |
| | | 3 | 2.89 | 0.41-20.28 | 0.285 |
| | | 4 | 67.31 | 2.92-1552.55 | 0.009 |
| Item 12 | Gastro-intestinal | 1 | 1.58 | 0.45-5.51 | 0.471 |
| | | 2 | 3.12 | 0.74-13.21 | 0.122 |
| | | 3 | 2.21 | 0.13-37.33 | 0.583 |
| | | 4 | – | – | – |
| Item 14 | Gastro-intestinal | 1 | 2.23 | 0.74-6.79 | 0.156 |
| | | 2 | 0.39 | 0.07-2.16 | 0.279 |
| | | 3 | 0.37 | 0.07-1.82 | 0.219 |
| | | 4 | 0 | 0- | 0.995 |
| Constant | | | 0.01 | 0-0.71 | 0.035 |

*Reference category: 'Keep in contact'*

**Emotional symptoms**

*Decision: 'Stay at home'*

| Item | Aspect, description | Response options | Risk Ratio | 95% CI | p-value |
|---|---|---|---|---|---|
| Item 22 | Feeling at home | 1 | 1.54 | 0.89-2.66 | 0.120 |
| | | 2 | 0.60 | 0.32-1.10 | 0.099 |
| | | 3 | 0.13 | 0.04-0.39 | <0.001 |
| | | 4 | 0 | 0- | 0.985 |
| Constant | | | 1.20 | 0.85-1.69 | 0.297 |

*Decision: Hospital admission*

| Item | Aspect, description | Response options | Risk Ratio | 95% CI | p-value |
|---|---|---|---|---|---|
| Item 22 | Feeling at home | 1 | 0.68 | 0.17-2.75 | 0.590 |
| | | 2 | 1.29 | 0.41-4.01 | 0.665 |
| | | 3 | 2.31 | 0.78-6.81 | 0.130 |
| | | 4 | 10.72 | 3.18-36.13 | <0.001 |
| Constant | | | 0.13 | 0.06-0.28 | <0.001 |

*Reference category: 'Keep in contact'*

**Self-management**

*Decision: 'Stay at home'*

| Item | Aspect, description | Response options | Risk Ratio | 95% CI | p-value |
|---|---|---|---|---|---|
| Item 21 | Reason for call | 1 | 6.74 | 1.89-24.00 | 0.003 |
| | | 2 | 1.51 | 0.78-2.95 | 0.224 |
| | | 3 | 0.60 | 0.22-1.63 | 0.314 |
| | | 4 | 0.41 | 0.13-1.25 | 0.117 |
| Item 24 | Handling | 1 | 3.07 | 1.62-5.80 | 0.001 |
| | | 2 | 7.50 | 2.78-20.23 | <0.001 |
| | | 3 | 0 | 0- | 0.990 |
| | | 4 | 0 | 0- | 0.994 |
| Item 27 | Preferences management | 1 | 1.75 | 0.87-3.52 | 0.115 |
| | | 2 | 0.86 | 0.39-1.92 | 0.720 |
| | | 3 | 0.06 | 0.02-0.18 | <0.001 |
| | | 4 | 0 | 0- | 0.995 |
| Constant | | | 0.57 | 0.31-1.04 | 0.066 |

*(Continued)*

**Table 3.** (Continued)

| Item | Aspect, description | Response options | Risk Ratio | 95% CI | p-value |
|---|---|---|---|---|---|
| *Decision: 'Hospital admission'* | | | | | |
| Item 21 | Reason for call | 1 | 0.88 | 0.05-14.20 | 0.925 |
| | | 2 | 0.45 | 0.08-2.52 | 0.366 |
| | | 3 | 0.21 | 0.02-2.20 | 0.193 |
| | | 4 | 2.28 | 0.64-8.13 | 0.202 |
| Item 24 | Handling | 1 | 1.63 | 0.50-5.33 | 0.417 |
| | | 2 | 2.51 | 0.56-11.28 | 0.229 |
| | | 3 | 5.39 | 0.57-50.58 | 0.141 |
| | | 4 | 2.30 | 0.16-32.25 | 0.537 |
| Item 27 | Preferences management | 1 | 1.24 | 0.23-6.76 | 0.805 |
| | | 2 | 1.36 | 0.25-7.52 | 0.726 |
| | | 3 | 0.38 | 0.07-1.98 | 0.251 |
| | | 4 | 8.91 | 1.90-41.77 | 0.006 |
| Constant | | | 0.08 | 0.02-0.30 | <0.001 |
| *Reference category: 'Keep in contact'* | | | | | |
| **Resources** | | | | | |
| *Decision: 'Stay at home'* | | | | | |
| Age | | | 0.94 | 0.88-1.01 | 0.079 |
| Item 29 | Support | 1 | 0.72 | 0.39-1.35 | 0.310 |
| | | 2 | 0.50 | 0.36-34.43 | 0.283 |
| | | 3 | 1.99e-09 | 0- | 0.999 |
| | | 4 | 0.46 | 0.04-5.21 | 0.529 |
| Item 31 | Companion | 1 | 1.21 | 0.70-2.09 | 0.495 |
| | | 2 | 1.57 | 0.46-5.30 | 0.467 |
| | | 3 | 0 | 0- | 0.999 |
| | | 4 | – | – | – |
| Item 32 | Distance to facility | 1 | 2.08 | 1.14-3.79 | 0.017 |
| | | 2 | 1.16 | 0.51-2.65 | 0.727 |
| | | 3 | 0.28 | 0.03-2.50 | 0.252 |
| | | 4 | 0 | 0- | 1.000 |
| Constant | | | 4.18 | 0.46-37.93 | 0.204 |
| *Decision: 'Hospital admission'* | | | | | |
| Age | | | 0.01 | 0.90-1.14 | 0.823 |
| Item 29 | Support | 1 | 0.38 | 0.12-1.18 | 0.095 |
| | | 2 | 3.97 | 0.29-54.98 | 0.304 |
| | | 3 | 0.63 | 0- | 1.000 |
| | | 4 | 11.05 | 1.60-76.25 | 0.015 |
| Item 31 | Companion | 1 | 3.83 | 1.58-9.26 | 0.003 |
| | | 2 | 8.27 | 1.75-39.01 | 0.008 |
| | | 3 | 0 | 0- | 1.000 |
| | | 4 | – | – | – |
| Item 32 | Distance to facility | 1 | 1.21 | 0.40-3.69 | 0.736 |
| | | 2 | 1.53 | 0.39-6.00 | 0.546 |
| | | 3 | 3.74 | 0.64-21.83 | 0.142 |
| | | 4 | 6.95 | 0.94-51.64 | 0.058 |
| Constant | | | 0.05 | 0-2.91 | 0.152 |
| *Reference category: 'Keep in contact'* | | | | | |

**Table 4. Inclusion and exclusion of items for the definitive GebStart-tool.**

| Items | Description | Face-validity, relevance, mean (min-max)[1] | Variance/ distribution and missings | Cox-regression models | Multinominal regression decision | Final decision | Additional justification |
|---|---|---|---|---|---|---|---|
| **Item 1** | **Contractions** | **3.50 (3-4)** | | **Very high** | ✓ | **Included** | |
| Item 2 | Contractions | 3.50 (3-4) | | Medium | | Excluded | |
| Item 3 | Contractions | 3.00 (2-4) | | Very low | | Excluded | |
| **Item 4** | **Contractions** | **3.25 (2-4)** | | **Very high** | | **Included** | |
| **Item 5** | **Contractions** | **3.38 (1-4)** | | **High** | ✓ | **Included** | |
| Item 6 | Vaginal discharge | 3.63 (3-4) | | High | | Excluded | Decision item 6 or 7 |
| **Item 7** | **Vaginal discharge** | **3.75 (3-4)** | | **Low** | ✓ | **Included** | Decision item 6 or 7, management decision |
| **Item 8** | **Vaginal discharge** | **3.88 (3-4)** | | **Very high** | ✓ | **Included** | Safety |
| Item 9 | Sleep | 3.38 (3-4) | | Very high | | Excluded | Decision item 9 or 10 |
| **Item 10** | **Fitness** | **3.38 (2-4)** | | **Medium** | ✓ | **Included** | Decision item 9 or 10 |
| Item 11 | Fitness | 3.38 (2-4) | | Low | | Excluded | |
| **Item 12** | **Gastro-intestinal** | **3.63 (2-4)** | | **Medium** | ✓ | **Included** | Decision item 12 or 14 |
| Item 13 | Gastro-intestinal | 3.63 (2-4) | X | | | Excluded | |
| Item 14 | Gastro-intestinal | 3.75 (2-4) | | Very low | ✓ | Excluded | Decision item 12 or 14 |
| **Item 15** | **Foetal movement** | **3.88 (3-4)** | | **Low** | | **Included** | Decision item 15 or 16, security, S2 Table |
| Item 16 | Foetal movement | 3.63 (3-4) | | Low | | excluded | |
| **Item 17** | **Confidence** | **3.13 (1-4)** | | **Very high** | | **Included** | |
| Item 18 | Emotional state | 3.50 (2-4) | | Medium | | Excluded | |
| Item 19 | Safe at home | 4.00 (4-4) | | High | | Excluded | Similarity item 22 |
| Item 20 | Worrying | 3.63 (3-4) | | Medium | | Excluded | Similarity item 22 |
| Item 21 | Reason for call | 4.00 (4-4) | | Very low | ✓ | Excluded | Decision Item 21 or 27 |
| **Item 22** | **Feeling at home** | **4.00 (4-4)** | | **Very high** | ✓ | **Included** | Decision item 19 or 22, item 20 or 22 |
| Item 23 | Distraction | 3.25 (2-4) | | Very high | | Excluded | Decision item 23 or 24 |
| **Item 24** | **Handling contractions** | **3.75 (3-4)** | | **High** | ✓ | **Included** | |
| Item 25 | Management at home | 4.00 (4-4) | X | | | Excluded | High number of missings |
| Item 26 | Attitudes | 3.75 (3-4) | | Medium | | Excluded | Decision item 26 or 27 |
| Item 27 | Preferences management | 3.88 (3-4) | | Low | ✓ | Excluded | Similarity item 22 |
| **Item 28** | **Preparation** | **3.50 (3-4)** | | **Very high** | | **Included** | |
| **Item 29** | **Support** | **3.75 (3-4)** | | **Medium** | ✓ | **Included** | |
| Item 30 | Support | 3.63 (3-4) | X | | | Excluded | Low variance, decision item 29 or 30 |
| **Item 31** | **Companion** | **3.38 (2-4)** | | **Very high** | ✓ | **Included** | |
| **Item 32** | **Distance to facility** | **4.00 (4-4)** | | **Medium** | ✓ | **Included** | |

X = Not included in regression models.

✓ = included in model.

[1] 1 = not relevant at all to 4 = very relevant.

**Table 5. Final items of the GebStart-tool and their description.**

| Item | Description | Response range |
|---|---|---|
| Item 1 | Frequency of contraction | from no contraction up to every 3–5 minutes |
| Item 4 | Painfulness of contractions | from not at all up to very painful |
| Item 5 | Behaviour during contraction | from talks normally to screams during contraction |
| Item 7 | Duration of vaginal fluid discharge | from no discharge up to >24 hours |
| Item 8 | Nature of vaginal discharge | from no discharge up to heavy bleeding or greenish fluid |
| Item 10 | Feeling fit | from very fit up to very exhausted |
| Item 12 | Eaten last time | from just up to not eaten >24 hours |
| Item 15 | Feeling foetal movement | from very much up to verry little, not at all |
| Item 17 | Confidence about birth | from very confident up to not confident at all |
| Item 22 | Feeling at home | from well, like to stay home, up to unwell, do not want to stay home |
| Item 24 | Handling contraction | from very well/no contraction up to not well at all |
| Item 28 | Feeling well prepared for birth | from very well up to not well at all |
| Item 29 | Support at home | from a lot of support up to no support |
| Item 31 | Handling situation by companion | from very well up to not well at all |
| Item 32 | Distance to hospital | from <10 minutes up to >60 minutes |

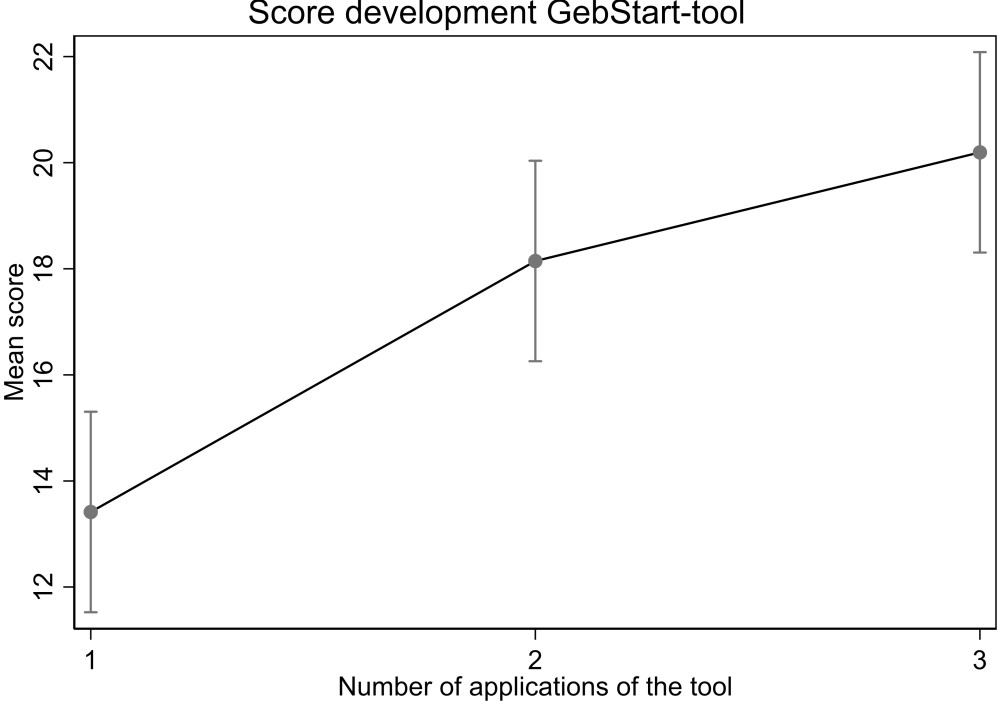

**Fig 2. Development of the total scores of women with three applications (n = 41).**

For several participants, the GebStart-tool was completed up to five times. The total score of participants with three completed tools (n = 41) showed a significant increase between the first, second and third completion in the Generalized Estimating equation (GEE) models (margin 13.41 for the first time, 18.15 for the second time and 20.20 for the third time, p < 0.001 for all comparisons, Fig 3). Sensitivity analysis excluding women with cervical dilatation > 6 cm showed a similar increase in the total score.

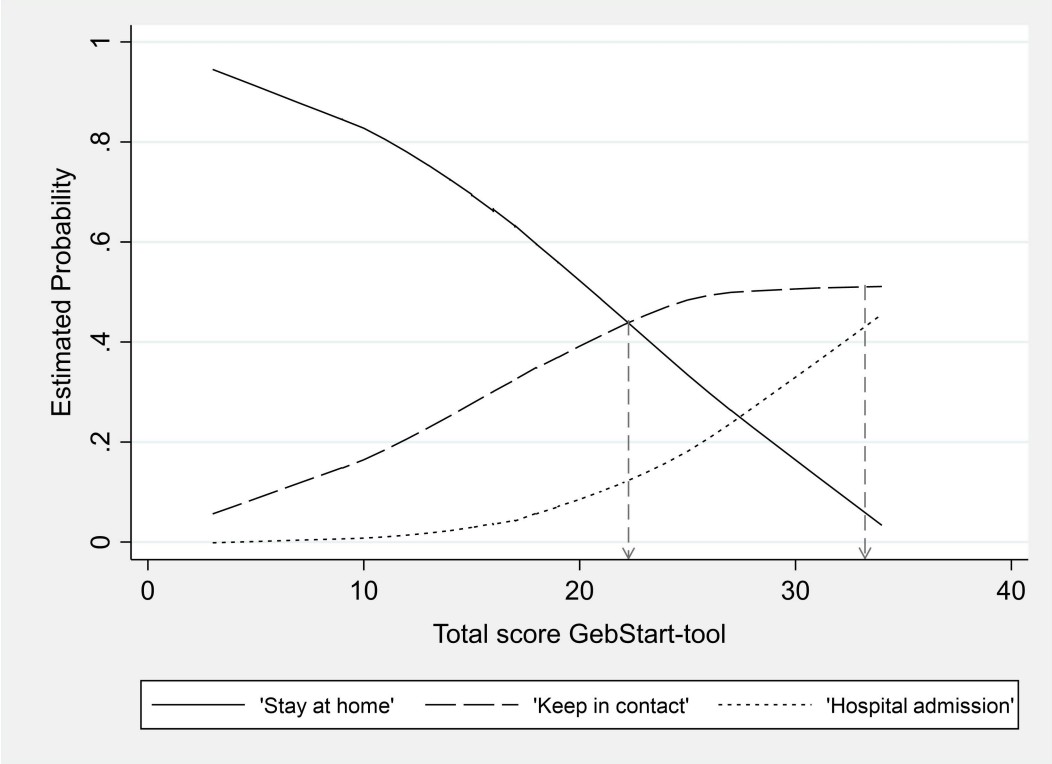

**Fig 3. Plotted expected probabilities for the final decision categories with respect to the total score of the GebStart-tool.** Arrows indicate the cut-off points calculated according to Bersabé & Riva [32].

Regarding convergent validity, a weak significant correlation between the CBSEI-score (15 labour items published by Schmidt et al. 2015) and item 17 assessing confidence (r=-0.12, p=0.029) and the CWS-score and item 17 (r=0.23, p<0.001) was found. There was no association between the CBSWI-score and item 22 assessing feeling at home (r=-0.06, p=0.292) and the CWS-score and item 22 (r=0.09, p=0.097).

On a scale from 0 to 10, women with a completed GebStart-tool evaluated their satisfaction about remaining at home, returning home and their overall experience with the onset of labour with a median of 7 each. A total of n=324 participants (90.5%) fully agreed that the midwife/doctor was friendly, n=282 (78.8%) that the midwife/doctor was empathic, n=296 (82.7%) that the midwife/doctor that the doctor gave a competent impression and n=275 (76.8%) that the midwife/the doctor responded to their needs.

## Discussion

In this study, we filled a gap in early labour care and developed and preliminary validated a tool to advise nulliparous women about the best place for them to stay during this labour phase. Items of the GebStart-tool have been reduced from 32 to 15. The total score of the reduced tool was significantly associated with all time intervals describing care needs (duration between completion of the tool and hospital admission as well as the first use of medical and alternative pain management), as well as with the care decision. Furthermore, a higher score of the reduced GebStart-tool was significantly associated with higher risk for the decision 'Hospital admission'.

The significant reduction of items was necessary due to the feedback from the clinicians at the study centres. A survey with midwives and doctors about their satisfaction with the application of the preliminary GebStart-tool, as well as from

focus group discussions with the midwifery teams showed that the length of the tool posed a major challenge in daily clinical practice due to time constraints [33].

When planning this study, we initially intended to use standard item reduction methods for assessing the emotional state of pregnant women in early labour [17]. However, additional aspects, such as attitude towards birth, antenatal preparation, support at home and the distance to the hospital, made the tool more complex than anticipated. The formative nature of the GebStart-tool became more pronounced with items about the emotional state in the minority. As a result, standard methods such as explorative factor analysis, were no longer suitable for item reduction and validation [18,30]. While challenges with items about physical symptoms were expected, the complexity of emotional and contextual aspects was unanticipated. Despite this, an exploratory factor analysis suggested four factors aligning with the domains identified during tool development [17,22,24,25]. Alternative analyses used for item reduction, including the PLS-SEM model, Cox regression models describing care needs and the multinomial logistic regression with the outcome care decision proved valuable alternatives. PLS-SEM models have been used in other studies to investigate the predictive value of constructs in complex, formative models [34] showed loadings of > 0.3 for most items in the current study. However, two items with loadings < 0.3 were retained due to their positive associations with care needs and care decision. These latter analyses were not described in the literature for item reduction and did not show a clear decision aid for all items. We therefore considered clinical aspects as well as the previously assessed face validity, which de Vet et al. [30] assessed as being very important for formative questionnaires.

Traditional validation approaches like confirmatory factor analysis and Cronbach's Alpha [30,35] were not applicable for this formative tool. Convergent validity was also limited, as only two items on emotional state could be included [17]. This suggests that the emotional state might be effectively assessed with fewer items. Additionally, aspects like birth preparation, attitude towards birth, support, distance to the hospital and practical usability in clinical settings were more important. The significant positive associations between the total score of the reduced GebStart-tool and the time intervals describing care needs and the care decision highlighted its suitability for assessing care needs. The inclusion of diverse aspects supported its effectiveness as a practical, multi-dimensional tool.

The final GebStart-tool with 15 items provides a potential range of 0–60 points. In our data, the observed range was 3–34 points in the fully completed tools. This means that the highest score reached was slightly more than half of the maximum possible points and that the cut-off points needed to be set rather low. We can only speculate why the total score was almost exclusively in the lower half of the range. The major reason seemed to be that the items of the tool covered many different aspects as described above, and women scored in some, but not in all of these aspects. It seemed important that not only women with painful contractions had higher care needs or a more urgent need for a decision to go to hospital, but also those without confidence, without support or living far from the hospital.

Reducing the GebStart tool from 32 to 15 items made it much more practical for clinical use while retaining key aspects beyond contractions and vaginal discharge. This is crucial, as some women in early labour have individual needs and report dissatisfaction with care and arbitrary decisions [1,10–12,25]. The GebStart tool holistically assesses women's conditions during early labour, considering not only physical discomfort but also well-being at home, women's confidence in giving birth, and handling contractions. This facilitates the identification of women who require professional support and no longer feel comfortable remaining at home, ensuring they receive appropriate care and are neither neglected nor dismissed [1,10]. Conversely, women who are comfortable at home should be encouraged to remain there for as long as possible, as later hospital admission has been associated with fewer interventions and improved obstetric outcomes [7,8]. This also enables the targeted use of resources for the women who need them [12]. The GebStart tool has the potential to support shared decision-making on the best place for each woman to stay. This is important for improving the provision of early labour care, but also women's experiences. However, further research is needed to evaluate the use and benefits of the shortened tool.

**Strengths and weaknesses**

A key strength of this study was the development of the GebStart-tool based on scientific evidence tailored to its purpose. Item generation was informed by focus group discussions on women's experiences and care needs during early labour [24,25] and the extensive scoping reviews [22,36]. The multicentre design with 627 participants in the main phase nearly 400 tool application was a notable achievement given the project's complexity. However, item reduction required separate analyses for the tool's four domains, using forward selection regression to avoid exceeding predictor limits [37]. This highlights a relatively tight sample size, which also only allowed for a preliminary validation of the tool. Additionally, the high education level of participants, coupled with other socio-demographic factors and the recruitment of women interested in early labour, limits the generalisability of findings. Some women may also have been in active labour (with advanced cervical dilatation) when completing the GebStart-tool.

The multicentre character of the study, reflecting the diversity of small maternity hospitals, allowed the tool's application in varied contexts. However, local adjustments in implementation occurred despite the provision of a study handbook, training sessions, and regular midwife meetings. Another limitation was the use of shortened CBSEI scale (15 items) published by Smidt et al. [27]. Although its reduction has been debated in the literature [27] the impact remains unclear. Additionally, the CBSEI and CWS were assessed at different gestational ages, possibly affecting convergent validity. Future studies could explore other scales, such as those assessing fear of childbirth. And finally, it must be noted that the second cut-off point for the decision 'Hospital admission' could be calculated but was not visible in the graph plotting the expected probabilities for the care decision categories with respect to the total score of the GebStart-tool.

**Conclusion**

Because of its formative and very complex character, items of the carefully developed, evidence-based GebStart-tool needed to be reduced, using alternative statistical analysis. This was done thoroughly, but not all analyses were used in previous studies. However, the output of this study was a practical instrument with 15 items, which could only be validated preliminarily due to the limited sample size and minor adjustments to two items. Further research is necessary for definitive validation of the GebStart-tool in larger and more diverse samples. Moreover, the use in clinical practice accompanied by implementation research and translation into other languages should be envisaged.

**Supporting information**

**S1 Fig. Scree plot of explorative factor analysis.** This scree plot supported the decision to subdivide the items into four domains.
(TIF)

**S1 Table. STROBE Statement—checklist of items that should be included in reports of observational studies.**
(DOCX)

**S2 Table. Cross loadings (>0.3) of items to the domains "Physical symptoms" "Emotional state" "Self-management" and "Resources" in the PLS-SEM model with the outcomes "Care needs" and "Care decision".**
(DOCX)

**S3 Table. Associations of the total score of the reduced GebStart-tool with the durations between the completion of the tool and hospital admission, onset of active labour, first medical and alternative pain management.**
(DOCX)

**S4 Table. Associations of the total score of the reduced GebStart-tool with the care decision 'Stay at home – Keep in contact – Hospital admission'.**
(DOCX)

**S1 File. Das GebStart-Tool.** Final German version of the GebStart-tool after item reduction, including 15 items.
(PDF)

**S2 File. Provisional English GebStart-Tool.** The GebStart-tool, preliminary, not validated version.
(PDF)

**S1 Checklist. Human Subjects Research Checklist.**
(PDF)

## Acknowledgments

Special thanks go to the midwifery teams in the participating study sites. Their support with completing the GebStart-tools during telephone calls with women during early labour was essential for the success of this study. We also thank all participating women for their time and effort to complete questionnaires and answer questions.

## Author contributions

**Conceptualization:** Susanne Grylka-Baeschlin, Antonia Nathalie Mueller.

**Data curation:** Susanne Grylka-Baeschlin, Nadine Pauli, Catherine Rapp, Carola Baumgartner, Clizia Iseppi, Nele Struebing, Linda Karg, Gabriela Minati, Leonhard Schäffer, Olav Lapaire, Markus Hodel, Gabriella Stocker, Nina Kimmich, Leila Sultan-Beyer, Antonia Nathalie Mueller.

**Formal analysis:** Susanne Grylka-Baeschlin, Antonia Nathalie Mueller.

**Funding acquisition:** Susanne Grylka-Baeschlin.

**Investigation:** Susanne Grylka-Baeschlin.

**Methodology:** Susanne Grylka-Baeschlin, Antonia Nathalie Mueller.

**Project administration:** Susanne Grylka-Baeschlin, Antonia Nathalie Mueller.

**Writing – original draft:** Susanne Grylka-Baeschlin, Antonia Nathalie Mueller.

**Writing – review & editing:** Nadine Pauli, Catherine Rapp, Carola Baumgartner, Clizia Iseppi, Nele Struebing, Linda Karg, Gabriela Minati, Leonhard Schäffer, Olav Lapaire, Markus Hodel, Gabriella Stocker, Nina Kimmich, Leila Sultan-Beyer.

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
