## [Decision Letter · Decision Letter 0]

10 Jun 2024

PONE-D-24-10925Development and validation of the GebStart-tool for advising primiparous women in early labourPLOS ONE

Dear Dr. Grylka-Baeschlin,

Thank you for submitting your manuscript to PLOS ONE. After careful consideration, we feel that it has merit but does not fully meet PLOS ONE’s publication criteria as it currently stands. Therefore, we invite you to submit a revised version of the manuscript that addresses the points raised during the review process.

Please note that we have only been able to secure a single reviewer to assess your manuscript. We are issuing a decision on your manuscript at this point to prevent further delays in the evaluation of your manuscript. Please be aware that the editor who handles your revised manuscript might find it necessary to invite additional reviewers to assess this work once the revised manuscript is submitted. However, we will aim to proceed on the basis of this single review if possible. 

The reviewer's comments are available below. They have requested a few clarifications about the Methods, including the sample size calculation. They have also asked that you provide further details about the utility of the model. Demonstrating the utility tool of a new method or tool is an essential criterion for publication in PLOS ONE, as is availability (https://journals.plos.org/plosone/s/submission-guidelines#loc-methods-software-databases-and-tools), so please ensure that you address this carefully as part of your revision, and also ensure you have shared a copy of the tool.

We look forward to receiving your revised manuscript.

Kind regards,

Marianne Clemence

Staff Editor

PLOS ONE

Journal Requirements:

2. PLOS ONE has specific requirements for studies that are presenting a new method or tool as the primary focus, including a newly developed or modified questionnaire or scale (https://journals.plos.org/plosone/s/submission-guidelines#loc-methods-software-databases-and-tools.) One requirement is that the questionnaire or scale must be openly available under a license no more restrictive than CC BY. In light of this, before we proceed, please include a copy of your questionnaire or scale as a Supporting Information file or provide a link if it is available through an online repository.

"This study was funded by the Swiss National Science Foundation with a Practice-to-Science Grant (PT00P1_199085)."

5. In the online submission form, you indicated that your data will be submitted to a repository upon acceptance.  We strongly recommend all authors deposit their data before acceptance, as the process can be lengthy and hold up publication timelines. Please note that, though access restrictions are acceptable now, your entire minimal  dataset will need to be made freely accessible if your manuscript is accepted for publication. This policy applies to all data except where public deposition would breach compliance with the protocol approved by your research ethics board. If you are unable to adhere to our open data policy, please kindly revise your statement to explain your reasoning and we will seek the editor's input on an exemption. 

Reviewers' comments:

Reviewer's Responses to Questions

**Comments to the Author**

1. Is the manuscript technically sound, and do the data support the conclusions?

Reviewer #1: Yes

2. Has the statistical analysis been performed appropriately and rigorously? 

Reviewer #1: Yes

3. Have the authors made all data underlying the findings in their manuscript fully available?

Reviewer #1: No

4. Is the manuscript presented in an intelligible fashion and written in standard English?

Reviewer #1: Yes

5. Review Comments to the Author

Reviewer #1: This is an interesting article, and I appreciate you working to fulfill an unmet need for advising patients in early labor.

Introduction: I'm a little challenged by the overall premise of the model. You report that patients should remain at home as long as possible to avoid unnecessary labor interventions, but for many patients laboring at home without sufficient support is unsatisfying. This is presented as a dichotomy, but could we improve the process of labor care such that we can support people in early labor without a decision tool?

Methods - can you please clarify planned sample size? For example you write that the target sample size was 550 participants assuming 20% dropout rate but that would target an analytic goal of 440 patients and I don't see in the paragraph where that number is explained.

I recognize the desire to maximize sample size, but given your tool is designed to determine which patients should present to the hospital inclusion of the patients who are in active labor is problematic with the adjustment in time to 0.001 hours. At very least, I think you need a sensitivity analysis re: stability of your results to this assumption.

Results - can you clarify what is meant by "median workload"? Is this equivalent to saying someone is a full time employee rather than someone who works a part time position?

I am not certain we can identify the correct threshold for hospital admission based on figure 3, since the percentage of patients recommended to 'keep in contact' is still higher.

Discussion: As a practicing clinician, I was left wondering about the role of this tool. The tool seems to perform pretty modestly at predicting admission / etc, and requires significant effort to complete as you note. I can imagine if I were to tell a patient that they should remain at home or come to the hospital because that's what most pregnant people chose to do under similar circumstances people responding that this is a case in which direct elicitation of individual preferences and shared decisionmaking would likely be faster and more fruitful.

6. PLOS authors have the option to publish the peer review history of their article (what does this mean? ). If published, this will include your full peer review and any attached files.

**Do you want your identity to be public for this peer review?** For information about this choice, including consent withdrawal, please see our Privacy Policy .

Reviewer #1: No

---

## [Author Response · Author response to Decision Letter 1]

9 Aug 2024

Please see as well the additional response letter!

Editorial comments

Thank you very much for this comment and the links. We adjusted the manuscript, tables, figures and supporting information accordingly.

2. PLOS ONE has specific requirements for studies that are presenting a new method or tool: the questionnaire or scale must be openly available under a license no more restrictive than CC BY. In light of this, before we proceed, please include a copy of your questionnaire or scale as a Supporting Information file or provide a link if it is available through an online repository.

We are happy to add the definitive German version of the GebStart-tool as Supporting Information and referenced it in the text:

S1 File. Das GebStart-Tool.

As the current version of the tool was de-signed to complete by the health care professional together with the parturient, we added on lines 359-361:

“The final German version of the GebStart-tool (S1 File) was designed to be completed by health professionals together with the parturients.”

Many thanks for spotting this discrepancy which we were happy to correct.

"This study was funded by the Swiss Na-tional Science Foundation with a Practice-to-Science Grant (PT00P1_199085)."

Thank you for proposing this amendment, which was added on lines 489-490 in the manuscript (with track-changes).

5. In the online submission form, you indi-cated that your data will be submitted to a repository upon acceptance. We strongly recommend all authors deposit their data before acceptance, as the process can be lengthy and hold up publication timelines. Please note that, though access re-strictions are acceptable now, your entire minimal dataset will need to be made freely accessible if your manuscript is ac-cepted for publication.

We are happy to share the data from the GebStart-Study simultaneously with the publication of this article. Therefore, we prepared and uploaded it with the related metafiles (code book and data processing information) to Zenodo (https://zenodo.org/). We uploaded all variables underlying this publication. Because of data protection, study site specific study IDs were replaced with consecutive numbers. All dates were omitted but the calculated time intervals were included in the data set. Additionally, some free text comments were deleted to remove information that could indicate persons or locations. Few variables (care models, Mother-Generated Index) were removed because of planned further publications and because data was not used for this publication. Data from n=28 participants of which n=10 had a completed tool, had to be removed because these women did not give consent for further data use. The data uploaded on Zenodo received the following doi-number: 0.5281/zenodo.12773347. Once the manuscript will be accepted, data will be openly accessible immediately.

We amended the manuscript as follows on lines 499-500:

“Data was uploaded to Zenodo and is available at the following link: 0.5281/zenodo.12773347.“

6. Your ethics statement should only ap-pear in the Methods section of your manuscript. If your ethics statement is written in any section besides the Methods, please delete it from any other section.

We are happy the delete the separate ethics statement at the end of the manuscript.

7. Please include captions for your Supporting Information files at the end of your manuscript, and update any in-text citations to match accordingly. Please see our Sup-porting Information guidelines for more information

Many thanks for this request. We added the captions in the Supporting Information paragraph on lines 462 and following up-dated as well the in text-citations.

Reviewer #1

1. Introduction: I'm a little challenged by the overall premise of the model. You report that patients should remain at home as long as possible to avoid unnecessary labor interventions, but for many patients laboring at home without sufficient support is unsatisfying. This is presented as a dichotomy, but could we improve the process of labor care such that we can support people in early labor without a decision tool?

Many thanks for this suggestion. We are happy to provide additionally explanations to describe the need of the study. We add-ed on lines 67-72 (manuscript with track changes) following sentences and adapted the subsequent sentence:

“However, especially in times of high work-load and financial pressure, advices might depend on the circumstances and can be arbitrary [11,12]. In addition, staying at home over 24 hours in women with prolonged early labour was also found to be associated with negative birth outcomes and an increased risk for caesarean section [13]. Therefore, this well-intended act to delay hospital admission…”

Furthermore, we added on lines 84-85 to improve the justification of the study:

“There is a lack of tools to assist early la-bour assessment in an evidence-based way to avoid arbitrary recommendations.”

2. Methods - can you please clarify planned sample size? For example you write that the target sample size was 550 participants assuming 20% dropout rate but that would target an analytic goal of 440 patients and I don't see in the para-graph where that number is explained.

We are grateful that you spotted this inconsistency. The estimated drop out was ap-prox. 25% and not 20% as described in the study protocol. We then rounded the target sample size up to n=550 participants. This was amended on lines 112-117.

3. I recognize the desire to maximize sam-ple size, but given your tool is designed to determine which patients should present to the hospital inclusion of the patients who are in active labor is problematic with the adjustment in time to 0.001 hours. At very least, I think you need a sensitivity analysis re: stability of your results to this assumption.

Thank you for this comment. All time points were assessed with minute precision and time intervals were therefore calculated also with minute precisions. Three decimals were only used in S2 Table for cross loadings in the PLS-SEM models. However, we agree that it is a limitation of the study that some women might have been in active labour at the time of completing the tool. We there-fore added in a further limitation on lines 439-441:

“Additionally, some women with advanced cervical dilatation at hospital admission might have been in active labour at the time of completing the GebStart-tool.”

Additionally, we performed sensitivity analysis for the validation of the study exclud-ing participants with cervical dilatations >6cm at hospital admission. The World Health Organization proposed the end of early labour at 5cm of cervical dilatation. As the GebStart-tool was mostly completed through telephone contact before hospital admission and because cervical dilatation could not be estimated at this time, 6cm appeared to be a reasonable threshold. We conducted a sensitivity analysis for the validation of the tool omitting all women with cervical dilatation >6cm at hospital admission. We added in the methods section the following paragraph on lines 201-208:

“As some women were admitted to the hospital in active labour, sensitivity analysis was performed for the Cox regression mod-els, the multinominal regressions, as well as the Generalised estimating equation (GEE) models used for the validation excluding women with cervical dilatation >6cm at the first vaginal examination after hospital ad-mission. The threshold of 6cm was decided because the World Health Organisation (WHO) defined the end of early labour at 5cm [4] and the cervical dilatation could not be assessed at the time of the completion of the tool. The median time interval between the first completion of the tool and hospital admission lasted 3.5 hours.”

In the result subchapter for the Validation (lines 316 and following), we referred to the sensitivity analysis several times.

4. Results - can you clarify what is meant by "median workload"? Is this equivalent to saying someone is a full time employee ra-ther than someone who works a part time position?

Thank you for this question. In Switzerland, people very often work part time. For this reason, it seems important to indicate the median of workload percentages. A 100% workload corresponds to a full-time em-ployment. We are happy to add a further footnote to Table 1:

“3 A 100% workload corresponds to full-time employ-ment and lower percentages to a part time position”

5. I am not certain we can identify the correct threshold for hospital admission based on figure 3, since the percentage of patients recommended to 'keep in contact' is still higher.

We agree that the second cut-off point is not self-explanatory, because there is no in-tersect between the curves on the graph. However, the cut-off points could also be calculated according to the suggestions of Bersabé & Riva (2010). We added this explanation to the heading of the figures. Additionally, we amended the explanations for the graph in the text on lines 349-350:

“However, for the second cut-off point, the plotted curves did not intersect, even though the values could be calculated according to the suggestions by Bersabé & Riva [32]..”

And finally, we added a further limitation on lines 450-453:

“And finally, it must be noted that the sec-ond cut-off point for the decision ‘Hospital admission’ could be calculated but was not visible in the graph plotting the expected probabilities for the final decision categories with respect to the total score of the GebStart-tool.”

6. Discussion: As a practicing clinician, I was left wondering about the role of this tool. The tool seems to perform pretty modestly at predicting admission / etc, and requires significant effort to complete as you note. I can imagine if I were to tell a patient that they should remain at home or come to the hospital because that's what most pregnant people chose to do under similar circumstances people responding that this is a case in which direct elicitation of individual preferences and shared deci-sion making would likely be faster and more fruitful.

Thank you for pointing out the missing discussion of the usefulness of the GebStart-tool for clinical practice. We are happy to add a paragraph in the discussion on lines 420 and following:

“The significant reduction from 32 to 15 items for the final GebStart-tool enabled to provide a much more manageable tool for clinical practice than it was possible during the study situation. Nevertheless, many as-pects were retained, allowing the decision for or against hospitalisation to depend on more than just contractions and vaginal discharge. This is important, because yet some pregnant women in early labour have individual needs and complain about unsatisfactory care and arbitrary decisions [1,10–12,25]. The application of the GebStart-tool has the potential to support shared decision-making about the best place to stay for every induvial parturient. However, further research is necessary to investigate the application of the shortened tool and its benefits.”

---

## [Decision Letter · Decision Letter 1]

25 Nov 2024

PONE-D-24-10925R1Development and validation of the GebStart-tool for advising primiparous women in early labourPLOS ONE

Dear Dr. Grylka-Baeschlin,

Thank you for submitting your manuscript to PLOS ONE. After careful consideration, we feel that it has merit but does not fully meet PLOS ONE’s publication criteria as it currently stands. Therefore, we invite you to submit a revised version of the manuscript that addresses the points raised during the review process.

We look forward to receiving your revised manuscript.

Kind regards,

David Desseauve, MD, MPH, PhD

Academic Editor

PLOS ONE

Reviewers' comments:

Reviewer's Responses to Questions

**Comments to the Author**

1. If the authors have adequately addressed your comments raised in a previous round of review and you feel that this manuscript is now acceptable for publication, you may indicate that here to bypass the “Comments to the Author” section, enter your conflict of interest statement in the “Confidential to Editor” section, and submit your "Accept" recommendation.

Reviewer #1: All comments have been addressed

Reviewer #2: (No Response)

Reviewer #3: (No Response)

2. Is the manuscript technically sound, and do the data support the conclusions?

Reviewer #1: Yes

Reviewer #2: Partly

Reviewer #3: Yes

3. Has the statistical analysis been performed appropriately and rigorously? 

Reviewer #1: Yes

Reviewer #2: Yes

Reviewer #3: Yes

4. Have the authors made all data underlying the findings in their manuscript fully available?

Reviewer #1: Yes

Reviewer #2: Yes

Reviewer #3: No

5. Is the manuscript presented in an intelligible fashion and written in standard English?

Reviewer #1: Yes

Reviewer #2: No

Reviewer #3: Yes

6. Review Comments to the Author

Reviewer #1: I appreciate the detailed response to my prior comments and have no further comments at this time.

Reviewer #2: TITLE

These are not primiparous women – see below

Same for the abstract

In the abstract conclusion I think that the term ‘evidence-based’ is somewhat confusing and disingenuous. The development methodology may be so, but using this term indicates that the tool itself has been the subject of a prospective interventional study or part of a systematic review, and this is clearly not case. Either change the phrase to be more precise or remove it entirely.

INTRODUCTION

I agree with the rational for an objective early labour assessment tool and the issues related to early or late presentation at the hospital.

This section is well written and creates a cogent and coherent argument for the study.

MATERIALS AND METHODS

• Typo : line 95 hospitals (plural)

• Terminology issues

Primiparous = someone who has given birth to one child

Primigravida = someone who is pregnant for the first time

Nulliparous = someone who has never had a live birth (not someone who has never been pregnant e.g. abortion, miscarriage)

You cannot use primiparous and primigravid interchangeably, as they are NOT the same.

• Understanding the methodology – very difficult due to the complexity of the steps required to arrive at the definitive 19 item tool

• Very difficult to understand what the plan was and how it was executed.

Unlike as it written in the BMJ Open protocol, it is unclear at which point and how the tool was implemented; and by whom. I suggest that this information is added as written in the published protocol (see below)

Data collection section of BMJ Open article:

‘The preliminary newly developed tool for the structured assessment is applied at the first contact with the parturient during early labour, which can either be by telephone or face to face. Midwives complete the questionnaire by asking the questions orally’

This is, in contrast, very clear

• I understand why the CBSEI and the CWS were used to validate the new tool, but it seems that neither has an exceptionally high sensitivity for objectively assessing confidence to cope with labour and nor accurately evaluating concerns about labour. Were there no other options? Also, it states that German language versions were used but were these translated versions also correctly validated prior to inclusion in the study? Each newly translated questionnaire requires validation prior to clinical use.

• These questions are very important as these methods were fundamental to the validation of the GebStart tool.

• Line 127: typo – mandatory (missing y)

• Line 129: ‘accidentally’ only the first subscale – this means what exactly? Full explanation in the text please.

• Line 164: can you do this and keep the data relevant? The Cox Regression excludes negative time intervals for good reason.

• Line 177: I find the term ‘outcome effective management unclear. Could this be changed or phrased better?

Do the authors know, on average, how long it took for the study midwives/clinicians to complete the 32-item questionnaire?

Someone who is anxious, and in discomfort is probably not going to want to spend too much time completing a questionnaire about their current condition. They want a fairly rapid response to their question: stay at home or come to hospital.

Presumably a 15-item questionnaire would take about half as long to get through.

RESULTS

Line 216 The term dropouts (37%) suggest a voluntary withdrawal from the study, however the context in which this is used is more of an exclusion. For example, a woman who was induced or underwent caesarean before the onset of labour. These were recruited women who were EXCLUDED but they were not ‘dropouts’ in the classic sense

For 15% of the 37% of dropouts (92 of 233) it is stated that the tool was not completed: reasons = forgotten, too much work or admission was urgent. BUT in the protocol, the tool was to be applied at the onset of labour, so for planned inductions and caesareans before the onset of labour, these patients did not satisfy the inclusion criteria to complete the GebStart Tool. Correct? So how could it have been forgotten when the patients were never suitable?

Characteristics of the population

The population is highly selective and would not be representative of populations in other similar European countries in terms of marital status, educational attainment, full-time employment, financial status, medical insurance and antenatal preparation. This is not a typical large (university) hospital population, so the results are unlikely to be externally valid.

These characteristics are also likely to suggest better physical and psychological preparation and coping strategies for labour, which may produce results considered as outliers when compared to more standard maternity populations.

Line 231 cannot be correct. If all participants were primiparous (one previous liveborn) how could 82% be in their first pregnancy? (primigravid) The terminology is incorrect throughout and must be amended. You cannot be both primiparous AND primigravid. This means that the terminology must be consistent throughout the article.

Primigravid or nulliparous but not primiparous. Please amend to provide consistency in the article.

Participants have been defined as ‘women expecting their first child’.

So nulliparous OR primigravid please but not primiparous.

Item reduction

This section is quite clear and well written despite being a rather complex process.

Validation

Should not the validation of the FINAL tool come after the section on cut-off points and designing the final versions?

This is a bit confusing made more so by the terminology and use of final Gebstart-tool for two different time points.

Is the final GebStart-tool mentioned in the validation section Line 310-311, the same final Gebstart-tool mentioned in Lines 351-360? I believe that this cannot be true.

Validation was a mathematical/comparative validation (using amongst other the CBSEI + CWS scores) rather than a prospective clinical validation of the final tool. I assume that there is a plan to prospectively evaluate the FINAL tool in clinically diverse populations, to confirm or refute the findings.

Could you also explain the difference between full CBSEI scale and a partial scale? Is a partial scale as valid as a full one? Please clarify in the text as this was an error it seems. And does this not weaken the assessment of the validation?

Cut-off points for the final instrument

The cut-off points are explained and the lack of intersect for the second-one, justified by the Bersabé and Riva reference. However, I think that given this anomaly, and the importance of the cut-off to the clinical use of the tool, the authors need to give more detail as to why they feel that this calculation (statistical fudge) is valid in these circumstances. Just citing reference is insufficient for me.

This is discussed in the strengths and limitations, but its significance is not explained.

Designing the final version

Just to be sure, it was not this final version with15 items and described modifications that was prospectively trialed clinically, rather the full 32 item list that was then reduced. This was more of a mathematical validation based on the patient responses to the 15 items selected from the original 32, aided by the CWS and partial CBSEI scoring systems to provide some alternative indirect clinical validity.

It seems that if the items were re-phrased based on the feedback of study centers, then the true final version of GenStart-tool is yet to be trialled/studied?

This is not very clear and needs to be clarified

DISCUSSION

Line 364 – primigravid NOT primiparous.

I would rephrase this whole paragraph as the it is the most important at summarising the findings of the study. In this form, the findings are not very clear.

‘Effective management decision’ – this term is a quite ambiguous in isolation. Why effective and not just management decision? Stay at home, stay in contact or go to hospital?

The output of the study was indeed a short GebStart tool but to state at this stage, without real world evaluations in diverse populations, that it is practical, remains to be confirmed.

Line 417-418: The results OF not TO the GebStart-tool should demonstrate this? Does this mean demonstrated in future clinical studies?

Strengths and weaknesses

I am not sure that the readers will correctly interpret ‘evidence-based’ as described. I assume that the authors refer to an ‘evidence-based methodology’ and nothing else.

They need to be very careful referring to ‘evidence-based’ as many will assume that this relates to the clinical use of the tool rather than its development. It would be quite a stretch to consider the GebStart tool ‘evidence-based at such an early stage of development.

Also, the study population is certainly not like that of any hospital where I have ever worked, and I have been lucky enough to have worked in several countries on 3 different continents. This makes external validity of the tool in other regions/countries essential if you are to prove its effectiveness. Currently it seems that it is designed for wealthy, well-educated, well insured and well supported married women. This type of population in most other countries is only found in the private sector.

These women are more likely to be better prepared for birth, physically and mentally stronger than most women that are seen in other larger European units.

As the tool attempts to evaluate coping and confidence in early labour, I would argue that the results of this study are only valid for this specific and unique group of women.

The discussion is too long and as with other sections, is not easy to follow.

I would suggest a complete re-write and then ask for feedback from some native English speakers, to confirm that it is relatively easy to understand.

CONCLUSION

I agree with the authors that more work is needed. At this stage the tool has been implemented in very small numbers in a very specific population of pregnant women, who are not at all representative of most women attending larger, university maternities in other European countries.

Reviewer #3: The aim of the study is interesting. The scientific answer is rather well constructed, with a coherent statistical basis. But I wonder about the selection bias, which I think reduces the external validity of this study. Indeed, patients who agree to be included are less worried about staying at home, which is not necessarily the case for a large part of the population. This type of patient will not come to hospital if the midwife is reassured that she is not in labour.

7. PLOS authors have the option to publish the peer review history of their article (what does this mean? ). If published, this will include your full peer review and any attached files.

**Do you want your identity to be public for this peer review?** For information about this choice, including consent withdrawal, please see our Privacy Policy .

Reviewer #1: No

Reviewer #2: No

Reviewer #3: No

---

## [Author Response · Author response to Decision Letter 2]

9 Jan 2025

The reviewer’s comments were very much appreciated. You will find the details about how we addressed each comment below in the ‘Revision2_Response to the reviewers’ table.

---

## [Editor Report · Decision Letter 2]

27 Jan 2025

PONE-D-24-10925R2Development and preliminary validation of the GebStart-tool for advising nulliparous women in early labourPLOS ONE

Dear Dr. Grylka-Baeschlin,

Thank you for submitting your manuscript to PLOS ONE. After careful consideration, we feel that it has merit but does not fully meet PLOS ONE’s publication criteria as it currently stands. Therefore, we invite you to submit a revised version of the manuscript that addresses the points raised during the review process.

The study is well-conducted and relevant. However, I suggest the following improvements:

Expand the discussion on the tool's clinical utility.Emphasize the preliminary nature of the findings due to sample size limitations.

With these revisions, the manuscript will be ready for publication.

We look forward to receiving your revised manuscript.

Kind regards,

David Desseauve, MD, MPH, PhD

Academic Editor

PLOS ONE
---

## [Author Response · Author response to Decision Letter 3]

12 Mar 2025

Please refer to the uploaded document 'Response to the editor'.

---

## [Editor Report · Decision Letter 3]

16 Mar 2025

Development and preliminary validation of the GebStart-tool for advising nulliparous women in early labour

PONE-D-24-10925R3

Dear Dr. Grylka-Baeschlin,

We’re pleased to inform you that your manuscript has been judged scientifically suitable for publication and will be formally accepted for publication once it meets all outstanding technical requirements.

Kind regards,

David Desseauve, MD, MPH, PhD

Academic Editor

PLOS ONE
---

## [Editor Report · Acceptance letter]

PONE-D-24-10925R3

PLOS ONE

Dear Dr. Grylka-Baeschlin,

I'm pleased to inform you that your manuscript has been deemed suitable for publication in PLOS ONE. Congratulations! Your manuscript is now being handed over to our production team.

Kind regards,

on behalf of

Dr. David Desseauve

Academic Editor

PLOS ONE